# Repair of G1 induced DNA double-strand breaks in S-G2/M by alternative NHEJ

Wei Yu[1], Chloé Lescale [1,5], Loelia Babin[2,5], Marie Bedora-Faure[1], Hélène Lenden-Hasse[1], Ludivine Baron[3], Caroline Demangel[3], José Yelamos [4], Erika Brunet[2] & Ludovic Deriano [1✉]

The alternative non-homologous end-joining (NHEJ) pathway promotes DNA double-strand break (DSB) repair in cells deficient for NHEJ or homologous recombination, suggesting that it operates at all stages of the cell cycle. Here, we use an approach in which DNA breaks can be induced in G1 cells and their repair tracked, enabling us to show that joining of DSBs is not functional in G1-arrested XRCC4-deficient cells. Cell cycle entry into S-G2/M restores DSB repair by Pol θ-dependent and PARP1-independent alternative NHEJ with repair products bearing kilo-base long DNA end resection, micro-homologies and chromosome transloca- tions. We identify a synthetic lethal interaction between XRCC4 and Pol θ under conditions of G1 DSBs, associated with accumulation of unresolved DNA ends in S-G2/M. Collectively, our results support the conclusion that the repair of G1 DSBs progressing to S-G2/M by alter- native NHEJ drives genomic instability and represent an attractive target for future DNA repair-based cancer therapies.

[1] Genome Integrity, Immunity and Cancer Unit, Equipe Labellisée Ligue Contre Le Cancer, Institut Pasteur, 75015 Paris, France. [2] Genome Dynamics in the Immune System Laboratory, Equipe Labellisée Ligue Contre Le Cancer, INSERM UMR 1163, Université Paris Descartes Sorbonne Paris Cité, Institut Imagine, 75015 Paris, France. [3] Immunobiology of Infection Unit, INSERM U1221, Institut Pasteur, 75015 Paris, France. [4] Cancer Research Program, Hospital del Mar Medical Research Institute, 08003 Barcelona, Spain. [5] These authors contributed equally: Chloé Lescale, Loelia Babin. ✉email: ludovic.deriano@pasteur.fr

DNA double-strand breaks are common lesions that continually threaten genomic integrity and must be faithfully repaired to prevent immunodeficiency, neurodegeneration, cancer and other diseases[1]. Two main pathways are used to repair DSBs and maintain genome integrity: homologous recombination (HR), which requires a sequence-homologous partner that is the homologous sister chromosome in S or G2 phase cells and non-homologous end-joining (NHEJ), which ligates the DNA ends without requiring extended homologies and is active throughout the cell cycle, particularly in G0/G1[2]. NHEJ is initiated by the binding of the Ku heterodimer to the broken DNA ends, creating a scaffold for the recruitment of other factors, including DNA-PKcs, XRCC4-ligase IV-XLF, PAXX, Artemis, and DNA polymerases[3]. Given that most mammalian somatic cells reside in G0/G1, NHEJ represents the major pathway for DSB repair and genome maintenance. However, NHEJ can also lead to chromosomal rearrangements in the context of complex DSB end structures or environments in which breaks are not resolved quickly[4,5]. This might be especially true in cells harboring dysfunctional p53 as the activity of the tumor suppressor gene *TP53* promotes G1/S cell cycle arrest and apoptosis in response to DNA breaks[6]. Consistent with this, deletion of p53 increases the overall frequency of translocations in cells with DSBs and complex chromosomal rearrangements are often found in tumors with p53 loss[7–11].

A third, less-well elucidated pathway termed alternative NHEJ (alt-NHEJ) has initially been described in cells with genetic deficiencies for one or more factors critical for NHEJ (e.g., XRCC4, Lig4, Ku70/80)[12–19]. Alt-NHEJ involves annealing of micro-homologies (MHs) before joining, is associated with excessive deletions and insertions at junction sites and has been implicated with the formation of large-scale genome rearrangements including chromosomal translocations[8,20]. Direct evidence that alt-NHEJ is error prone on a genome-wide scale came from the analysis of NHEJ-deficient mice that are also deficient for p53[20–23]. Ku80/p53 or XRCC4/p53-doubly deficient mice lack mature lymphocytes because the NHEJ/p53-deficient lymphocyte progenitors cannot efficiently assemble and express functional immunoglobulin (Ig) and T cell receptor (TCR) genes needed to drive expansion and development. Nevertheless, these animals invariably develop pro-B cell lymphomas harboring oncogenic chromosomal translocations involving the Ig heavy chain (*Igh*) and Myc loci, all of which harbor MHs at the breakpoints[20–23].

Alt-NHEJ depends on factors (e.g., CtIP, MRN, EXO1) that resect DSBs to expose single-stranded DNA (ssDNA) overhangs at sites of DSBs and relies on the activity of DNA polymerase theta (Pol θ, encoded by *Polq* in mice) that promotes annealing of ssDNA containing MHs and completes DNA synthesis to fill in the resected gap before ligation terminates the repair. Alt-NHEJ may also include Poly-(ADP-ribose)-polymerase (PARP) 1 that catalyzes the poly-(ADP-ribosylation) of proteins at DSB sites and may provide DNA end tethering or protein scaffolding activities necessary for the end-joining reaction[24–30]. The relative contribution of Pol θ and PARP1 to the formation of chromosomal translocations and whether they work together in alt-NHEJ is unclear[25]. In addition, the efficacy of alt-NHEJ during the different phases of the cell cycle remains to be examined. Indeed, while (micro)-homology usage and DNA end resection are features of alt-NHEJ that are consistent with a prevalence for this pathway in S/G2[2], the observation that alt-NHEJ serves as a backup for both NHEJ (e.g., in cells deficient for Ku70/80 or XRCC4/Lig4) and HR (e.g., in cells deficient for BRCA1/BRCA2) indicates that it might be active throughout the cell cycle[31–33]. To investigate these questions, we develop an experimental approach in which DNA DSBs can be induced in G1-arrested cells and their repair tracked in G1 and upon cell cycle entry into S-G2/M. We apply cytogenetics and high-throughput sequencing assays to measure end joining in a panel of mouse pro-B cell lines deficient for NHEJ (XRCC4), alt-NHEJ (PARP1 and Pol θ) and the G1/S cell cycle checkpoint p53. We show that in XRCC4/p53-doubly deficient cells, joining of G1-induced DNA breaks occurs in S-G2/M and leads to extensive genetic instability with repair products bearing kilo-base long DNA end resection, micro-homologies and chromosome translocations. We find that such repair events are independent of PARP1 and rely on Pol θ that enables the survival and proliferation of XRCC4/p53 cells exposed to G1 DSBs by limiting the accumulation of unresolved DNA ends in mitosis. Our results shed light and provide mechanistic insight into a previously underestimated DNA damage repair event—the repair of G1-induced DSBs in the subsequent S-G2/M phase of the cell cycle—that likely contributes to genetic instability in cancer cells and represents a promising therapeutic target.

## Results

**Alt-NHEJ rescues RAG-induced recombination in S-G2/M.** We employed CRISPR/Cas9-mediated gene editing to delete *Xrcc4* (Δexon3) from Abelson kinase (*v-Abl*) transformed BCL2-expressing wild type (WT) *v-Abl* progenitor (pro)-B cells, generating *Xrcc4*$^{-/-}$ *v-Abl* pro-B cell clones[34,35] (See the list of cell lines and primers used in this study in Supplementary Tables 1 and 2, respectively, as well as the genotyping and Western blot analyses of these cell lines in Supplementary Fig. 1). Treatment of *v-Abl* pro-B cells with a *v-Abl* kinase inhibitor (STI571, hereafter referred to as ABLki) leads to G1 cell cycle arrest, the rapid induction/stabilization of RAG1/2 gene expression and rearrangement of the endogenous *Igk* locus or any introduced V(D)J recombination reporter substrate[34–37] (Fig. 1a). We transduced *v-Abl* pro-B cells from each genotype with the pMX-RSS-GFP/IRES-hCD4 retroviral recombination substrate in which GFP is expressed upon successful chromosomal inversional RAG-mediated recombination (Supplementary Fig. 2A)[34–36]. Flow cytometry analysis and PCR amplification of endogenous *IgkV$_{10-95}$-J$_4$* rearrangement revealed robust recombination in ABLki-treated/G1-arrested WT cells (Supplementary Fig. 2). In line with previous studies[34], we found that V(D)J recombination was severely compromised in *Xrcc4*$^{-/-}$ pro-B cell clones as assessed by flow cytometry (Supplementary Fig. 2A, B) and PCR amplification of endogenous *IgkV$_{10-95}$-J$_4$* rearrangement that revealed a complete lack of coding join formation in ABLki-treated *Xrcc4*$^{-/-}$ pro-B cells (Supplementary Fig. 2C, D). Because unrepaired RAG-induced DNA breaks trigger p53-dependent G1/S checkpoint activation[38,39], we used CRISPR/Cas9 editing to generate XRCC4/p53 doubly deficient cell clones from *p53*$^{-/-}$ *v-Abl* pro-B cell lines and assessed cell cycle re-entry and DSB repair in these settings. In contrast to *Xrcc4*$^{-/-}$ cells that remained blocked in G1 upon ABLki removal, WT, *p53*$^{-/-}$ and *Xrcc4*$^{-/-}$ *p53*$^{-/-}$ *v-Abl* pro-B cells readily re-entered the cell cycle following wash off of ABLki (Fig. 1a, b; Supplementary Fig. 3). PCR and flow cytometry analysis of ABLki-treated *Xrcc4*$^{-/-}$ *p53*$^{-/-}$ *v-Abl* pro-B cells revealed a severe V(D)J recombination defect in these cells (Fig. 1c; Supplementary Fig. 2B). PCR amplification analysis confirmed robust *IgkV$_{10-95}$-J$_4$* rearrangement in G1-arrested XRCC4-proficient pro-B cells—WT pro-B cells, *p53*$^{-/-}$ pro-B cells and *Xrcc4*$^{-/-}$ *p53*$^{-/-}$ pro-B cells complemented with XRCC4—with similar recombined products detected in released/cycling conditions (Fig. 1c). Strikingly, *IgkV$_{10-95}$-J$_4$* PCR amplification analysis revealed a substantial rescue of recombination in released *Xrcc4*$^{-/-}$ *p53*$^{-/-}$ pro-B cells in the form of PCR amplified DNA products of highly variable sizes (Fig. 1c). Topo-cloning and Sanger sequencing of these products revealed DNA sequences that located 5' of the *Jk$_4$* cutting site, suggesting extensive DNA end

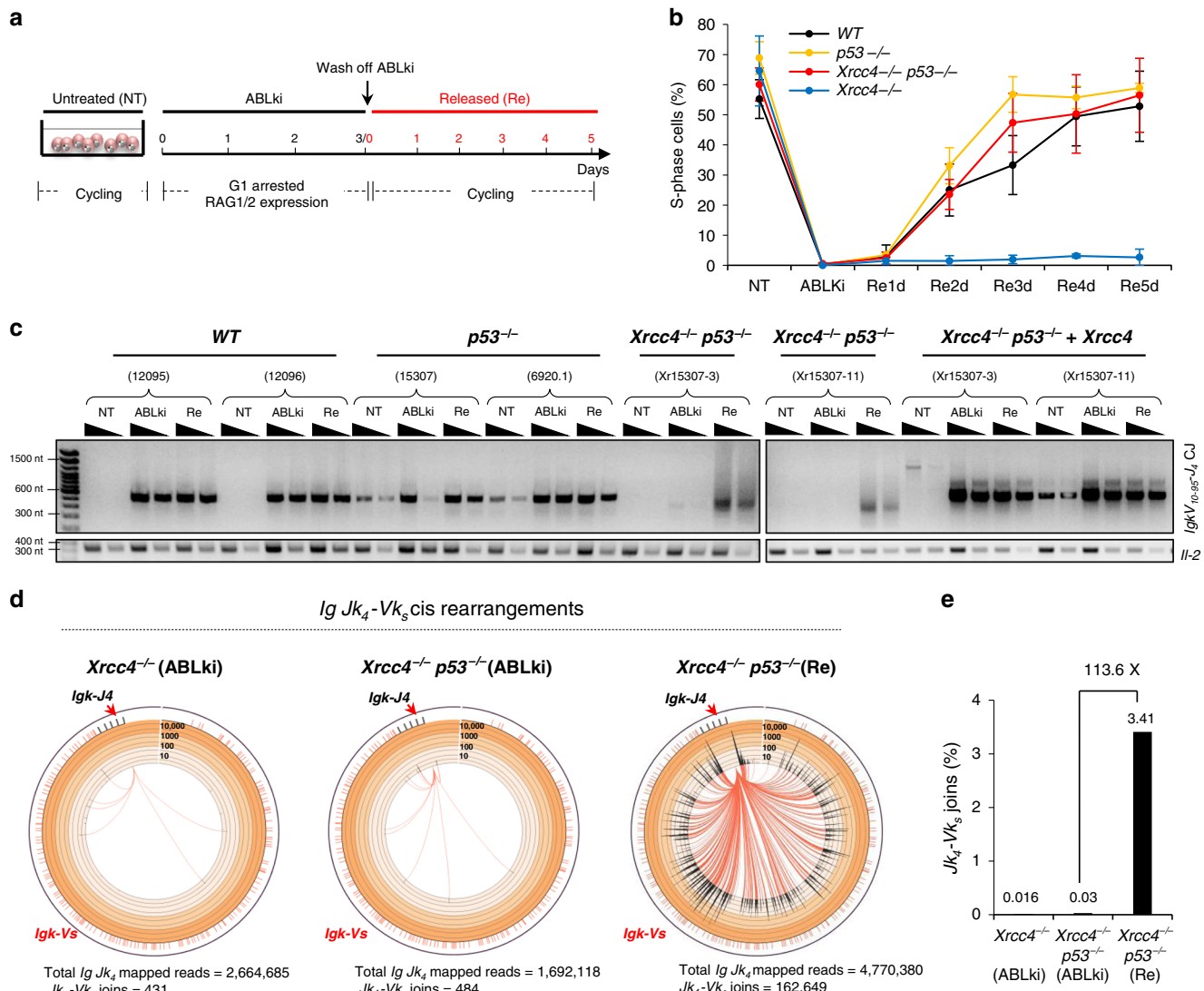

**Fig. 1 Alt-NHEJ rescues RAG-induced recombination in S-G2-M. a** Schematic of DSB induction by RAG endonuclease in G1-arrested *v-Abl* pro-B cells. Kinetics of ABLki treatment and release are indicated. **b** Quantification of the percentage of S phase cells (PI+/Edu+ cells) in untreated conditions (NT), after exposure to ABLki for 72 h (ABLki) and one to five days after washing off ABLki (Re1d to Re5d). Lines represent means ± S.D. with a minimum of two independent B cell clones for each genotype. The number of independent experiments for each genotype and time point is indicated in the source data file. **c** Representative semi-quantitative nested PCR analysis of *Igk_{V10-95}-J_4* coding join in untreated (NT), G1 blocked (ABLki) and released/cycling (Re) *v-Abl* pro-B cells of the indicated genotype. *Il-2* gene PCR was used as a loading control. Two independent cell lines were used for each genotype: WT (*12095* & *12096*); *p53^{-/-}* (*15307* & *6920.1*); *Xrcc4^{-/-} p53^{-/-}* (*Xr15307-3* & *Xr15307-11*). Source data are provided as a Source Data file. The PCR gel is representative of $n = 3$ independent experiments. **d** Circos plots displaying *IgkJ_4-IgkV_{region}* prey junction distribution from *IgkJ_4* coding end bait libraries. Junctions are represented as arcs originating from *IgkJ_4* breaks with a minimum of 5 reads per 1,000 bp bin for *Xrcc4^{-/-}* G1 arrested libraries, 4 reads per 1000 bp bin for *Xrcc4^{-/-} p53^{-/-}* G1 arrested libraries and 10 reads per 1000 bp bin for *Xrcc4^{-/-} p53^{-/-}* released libraries. Released cells were harvested 4 to 6 days after washing off ABLki. **e** Quantification of *IgkJ_4-V_{region}* junctions in *Xrcc4^{-/-}* and *Xrcc4^{-/-} p53^{-/-}* *v-Abl* pro-B cells. Values are the percentages of junctions relative to total mapped reads. Fold enrichment in *Xrcc4^{-/-} p53^{-/-}* released cells (Re) as compared to G1 blocked cells (ABLki) is indicated. Graph bars represent the pool of $n = 4$ independent experiments for *Xrcc4^{-/-}* G1 blocked cells, $n = 3$ for *Xrcc4^{-/-} p53^{-/-}* G1 blocked cells and $n = 6$ for *Xrcc4^{-/-} p53^{-/-}* released cells with two independent cell lines for each genotype. See also Supplementary Table 4.

resection from nearby *Jk_1* and *Jk_2* coding ends prior to joining in *Xrcc4^{-/-} p53^{-/-}* released pro-B cells, in addition to microhomology (MH) at breakpoint junctions (Supplementary Fig. 4; Supplementary Table 3).

To assess recombination in a more robust and extensive manner, we applied linear amplification-mediated high-throughput genome-wide translocation sequencing (LAM-HTGTS) that enables the mapping, at nucleotide resolution, of junctions between LAM-HTGTS "bait" DSBs and DSBs genome-wide[40].

We used RAG-initiated DSBs in the 3' *IgkJ_4* coding segment region as a bait DSB to capture joins to RAG-induced prey DSBs within the upstream *IgkV* locus region in XRCC4-proficient and XRCC4-deficient pro-B cell clones (Supplementary Fig. 5). Sequencing of four independent LAM-HTGTS libraries from ABLki-treated WT pro-B cells recovered 4,855,815 reads that mapped specifically to the *IgkJ4* bait region. Among those, 612,718 reads (12.61% of total *IgkJ4* mapped reads) corresponded to joins between *IgkJ_4* coding ends and the $V_k$ region

(Supplementary Fig. 6; Supplementary Table 4). Notably, in untreated cycling cells, $IgkJ_4$-$V_{region}$ joins represented less than 0.01 % of total $IgkJ4$ mapped sequences indicating that RAG-mediated $Igk$ recombination is specifically induced in ABLki-treated pro-B cells (Supplementary Table 4). Analysis of ABLki-treated $p53^{-/-}$ pro-B cells also revealed robust recombination levels (8.41%) in these cells (Supplementary Fig. 6; Supplementary Table 4). By sharp contrast, analysis of ABLki-treated $Xrcc4^{-/-}$ and $Xrcc4^{-/-}$ $p53^{-/-}$ pro-B cells revealed very low levels of recombination with 0.02% (431/2,664,685 reads) and 0.03% (484/1,692,118 reads) of total $IgkJ4$ mapped reads corresponding to $IgkJ4$-$V_{region}$ joins in these cells, respectively (Fig. 1d, e; Supplementary Table 4). Consistent with $IgkV_{10-95}$-$J_4$ rearrangement results (Fig. 1c), $IgkJ4$-bait LAM-HTGTS revealed rescue of recombination in released $Xrcc4^{-/-}$ $p53^{-/-}$ pro-B cells with more than 3% of total $IgkJ4$ mapped reads corresponding to $IgkJ4$-$V_{region}$ joins (162,649/4,770,380 reads) (Fig. 1d, e; Supplementary Table 4). We confirmed rescue of $Igk$ recombination in released $Xrcc4^{-/-}$ $p53^{-/-}$ pro-B cells using a locus-specific pulldown approach that enables the capture of small genomic regions flanking RAG-DSB sites at canonical $Vk$ and $Jk$ segments before high-throughput sequencing (Supplementary Fig. 7; Supplementary Table 4). Together, these results demonstrate that, in the absence of XRCC4, RAG-induced DSB repair is not functional in G1-arrested pro-B cells and substantially rescued upon release into the cell cycle.

**Alt-NHEJ of G1 DSBs in S-G2/M is error-prone**. To investigate the nature of the backup repair pathway that is operative in XRCC4/p53-deficient cells, we next examined breakpoint junctions at $IgkV$ RAG-cutting sites (i.e., $IgkV$ coding ends or $IgkV$ signal ends joined to $IgkJ_4$ bait coding ends) by assessing DNA end resection length and MH usage at these breakpoint sites (Supplementary Fig. 8A). To measure end resection events most accurately, we only analyzed annotated $IgkV$ recombining segments that localized more than 4 kb one from another. As expected, $IgkJ_4$-to-$V$ rearrangements from G1-arrested NHEJ-proficient pro-B cells contained minimal resection (mean resection = 3.5 bp in WT pro-B cells and 3.2 bp in $p53^{-/-}$ pro-B cells), harbored short nucleotide insertions generated by terminal deoxynucleotidyl transferase during processing of RAG-DNA ends (32.2% of the joins contained nucleotide insertions in WT pro-B cells, 18.2% in $p53^{-/-}$ pro-B cells) and contained relatively low levels of MH at breakpoint junctions (2-to-6 bp MH: 17.3% in WT pro-B cells, 21.6% in $p53^{-/-}$ pro-B cells) (Fig. 2a, b; Supplementary Fig. 8B–D). By contrast, rare $IgkJ_4$-to-$V$ rearrangements recovered from G1-arrested $Xrcc4^{-/-}$ pro-B cells lacked nucleotide insertions (7.5%), contained more frequently MHs (2-to-6 bp MH: 43.3%) and suffered increased DNA end-processing prior to joining (Fig. 2a, b; Supplementary Fig. 8B–D). This type of resection was nevertheless limited with a mean resection length of 7.5 nucleotides and less than 0.1% of the joins containing more than 100 nucleotides loss (Fig. 2a, b). The analysis of $IgkJ_4$-to-$V$ rearrangements recovered from released WT and $p53^{-/-}$ pro-B cells revealed similar breakpoint junction features as in G1-arrested conditions, consistent with the fact that RAG-induced DSBs in G1 cells were repaired prior to ABLki wash off and release into the cell cycle (Fig. 2a–c; Supplementary Fig. 8B–E). By sharp contrast, the analysis of $IgkJ_4$-to-$V$ rearrangements recovered from released $Xrcc4^{-/-}$ $p53^{-/-}$ pro-B cells revealed extensive DNA end resection with a mean resection length of 730.1 nucleotides and more than 62% of the joins containing over 100 nucleotides loss (Fig. 2a, c). The majority of these joins contained 2-to-6 nucleotides MH (58.5%) at their breakpoint junctions consistent with a repair by alt-NHEJ[24] (Fig. 2c;

Supplementary Fig. 8C, E). Together, these results show that in XRCC4/p53-deficient pro-B cells, cell cycle entry into S-G2/M phase promotes the repair of G1-induced DSBs through error-prone alt-NHEJ that is associated with kilo-base long nucleotide resection and MH usage.

**Unrepaired G1 DSBs promote chromosome translocations in S-G2/M**. Using HTGTS, we measured RAG-induced chromosomal translocations arising between the $IgkJ_4$ DSB bait and DSBs occurring within the chromosome 16-localized $Igl$ and chromosome 12-localized $Igh$ loci (Supplementary Fig. 5). Analysis of ABLki-treated WT and $p53^{-/-}$ pro-B cells revealed low levels of chromosomal translocations with 0.003% (127/4,855,815 reads) and 0.002% (40/2,045,431 reads) of total $IgkJ4$ mapped reads corresponding to $IgkJ_4$-to-$Igl/h$ joins in these cells, respectively (Supplementary Fig. 9; Supplementary Table 4). In cell cycle released conditions, $IgkJ_4$-to-$Igl/h$ chromosomal translocations remained at low levels in WT pro-B cells (0.002%; 96/6,196,042 reads) and increased approximately 18-fold in p53-deficient pro-B cells (0.035%; 765/2,162,548 reads) indicating that the p53 G1/S checkpoint restrains RAG-induced chromosomal translocation formation in NHEJ-proficient $v$-$Abl$ pro-B cells (Supplementary Fig. 9; Supplementary Table 4). $IgkJ_4$-to-$Igl/h$ chromosomal translocations were almost undetectable in G1-arrested $Xrcc4^{-/-}$ and $Xrcc4^{-/-}$ $p53^{-/-}$ pro-B cells with less than 0.0003% of total $IgkJ4$ mapped reads corresponding to $IgkJ_4$-to-$Igl/h$ translocations in these cells (3/2,664,685 and 4/1,692,118 reads, respectively) (Fig. 3a, b; Supplementary Table 4). By contrast, HTGTS analysis revealed a dramatic increase in chromosomal translocations (570-fold) in released $Xrcc4^{-/-}$ $p53^{-/-}$ pro-B cells with 0.132% (6,311/4,770,380 reads) of total $IgkJ4$ mapped reads corresponding to $IgkJ_4$-to-$Igl/h$ joins in these cells (Fig. 3a, b; Supplementary Table 4). These results demonstrate that unrepaired RAG-induced broken DNA ends generated in G1-phase cells promote chromosomal translocations at high frequency upon cell cycle entry into S phase.

We next assayed Cas9-dependent induction of DSBs for chromosomal translocation formation[5,41] using guide RNAs (gRNAs) targeting the $Abl$ locus on chromosome 2 and the breakpoint cluster region ($Bcr$) on chromosome 10, which represent physiological breakpoint sites (Fig. 3c and Supplementary Fig. 10A). We assessed chromosomal translocation formation in RAG2-deficient $v$-$Abl$ pro-B cell lines to prevent interference from the RAG nuclease. Importantly, this experimental system makes it possible to induce a DSB in cells stopped in G1 but also in proliferating cells (Fig. 3c). In agreement with previous studies[42], in cycling conditions, chromosomal translocation frequency was relatively low in NHEJ-proficient $p53^{-/-}$ cells (frequency $\sim 7.2 \times 10^{-3}$) and increased 4-fold in XRCC4/p53-deficient cells (frequency $\sim 2.9 \times 10^{-2}$) while retaining similar translocation junctions bearing deletions, insertions and micro-homologies (Fig. 3d–f; Supplementary Fig. 10B, C; Supplementary Table 5). Of note, analysis of nucleotide insertions and deletions (InDels) at $Abl$ and $Bcr$ Cas9 cutting sites revealed efficient cutting and joining activities in both cell types with longer resections observed in $Xrcc4^{-/-}$ $p53^{-/-}$ cycling pro-B cells (Fig. 3d; Supplementary Fig. 10D, E). In sharp contrast, Cas9-induced chromosomal translocations were virtually absent from G1-arrested $p53^{-/-}$ and $Xrcc4^{-/-}$ $p53^{-/-}$ pro-B cells (translocation frequency of $\sim 2.2 \times 10^{-4}$ and $\sim 2.1 \times 10^{-5}$, respectively) (Fig. 3d–e, Supplementary Fig. 10B). InDels levels were lower in G1-arrested $p53^{-/-}$ cells (4–26%) as compared to cycling $p53^{-/-}$ cells (25–89%) that harbored small deletions of up to 10 nucleotides (Fig. 3d; Supplementary Fig. 10D, E). This suggests that canonical NHEJ is largely error-free in G1-phase cells and

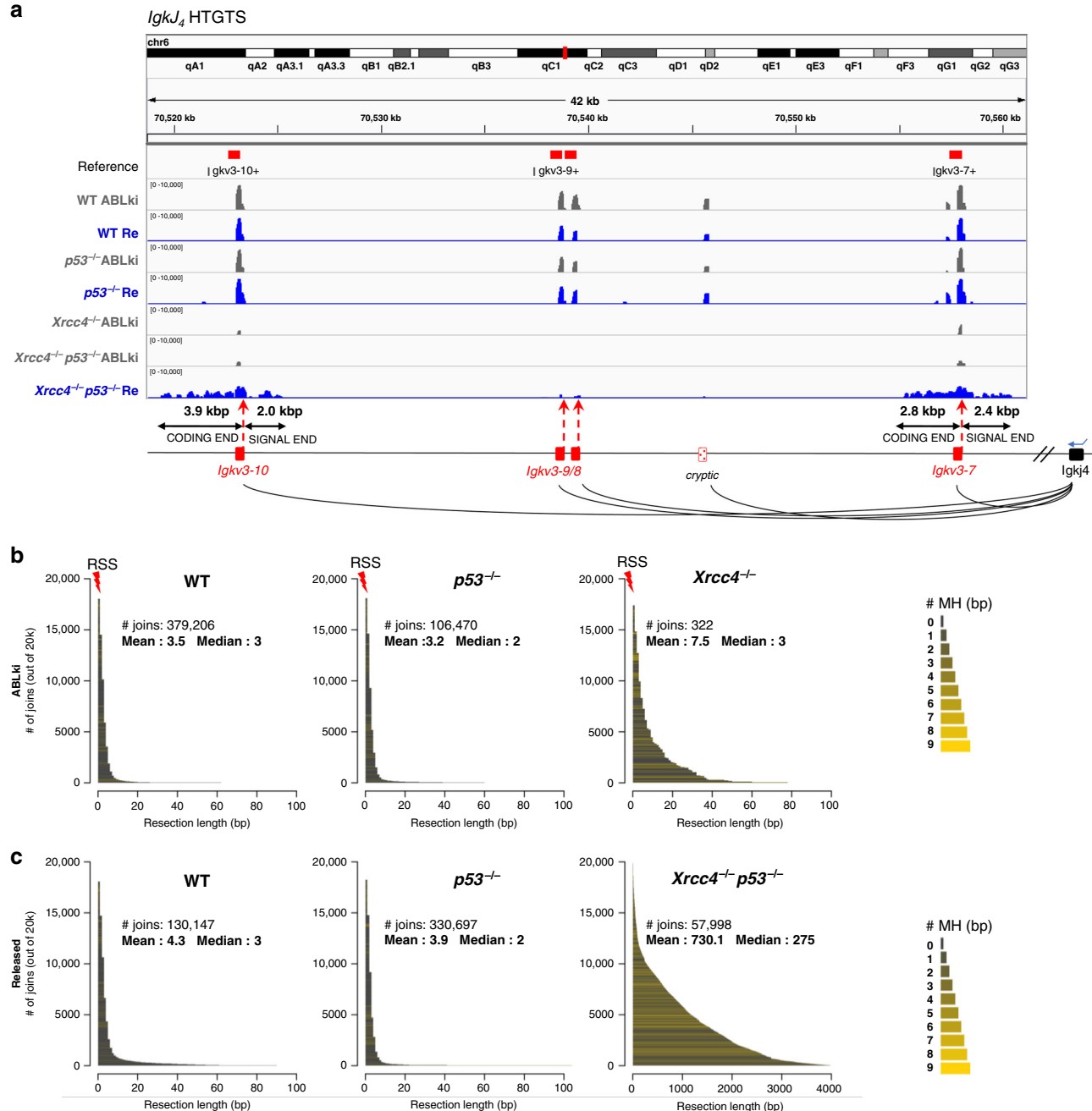

**Fig. 2 Alt-NHEJ of G1-induced DSBs in S-G2/M is associated with extensive DNA end resection and microhomology. a** IGV plot displaying representative $IgkJ_4$-$V_{region}$ junctions in G1 blocked (ABLki, gray) and released/cycling (Re, blue) v-Abl pro-B cells of the indicated genotype. $IgkV$ segments are represented by red boxes and recombination signal sites (RSS) are marked with red dash lines. Track height represents the number of junctions (log scale). Maximum height is set to 10,000 for all genotypes. **b, c** Bar graphs displaying the extent of resected DNA for each end joining product relative to the RSS for ABLki-treated cells (**b**) and released cells (**c**) of the indicated genotype. Bars are shaded according to the extent of microhomology as indicated. The height of each bar on the y axis defines the proportion of each product in a set of 20,000 estimated input molecules. To measure end resection events most accurately, only the annotated $IgkV$ recombining segments that localized more than 4 kb one from another were analyzed. The total number of analyzed joins and the mean and median resection length of junctions are indicated. Note the change of x axis scale for released $Xrcc4^{-/-}$ $p53^{-/-}$ cells. See also Supplementary Table 4.

more error-prone in S-G2/M. Notably, InDels were virtually absent from $Xrcc4^{-/-}$ $p53^{-/-}$ pro-B cells (0 to 3%) which is consistent with a severe end-joining defect in G1-arrested XRCC4-deficient cells. Strikingly, in released conditions, while $p53^{-/-}$ cells remained mainly translocation-free (frequency ~ 3 × $10^{-4}$), $Xrcc4^{-/-}$ $p53^{-/-}$ cells accumulated very high level of $Bcr/Abl$ chromosomal translocations (frequency ~ $2.7 \times 10^{-2}$) associated with a dramatic increase of InDels at the $Abl$ and $Bcr$

Cas9 target sites (75 to 100%) (Fig. 3d, e; Supplementary Fig. 10B–E). Further analysis of chromosomal translocation and InDels from $Xrcc4^{-/-}$ $p53^{-/-}$ released cells revealed extended deletions and microhomologies at junction sites (Fig. 3f and Supplementary Fig. 10D, E). Thus, consistent with results obtained from RAG-induced DNA breaks, these results demonstrate that unrepaired Cas9-induced G1 DSBs lead to genetic instability in the form of unfaithful repair in cis and

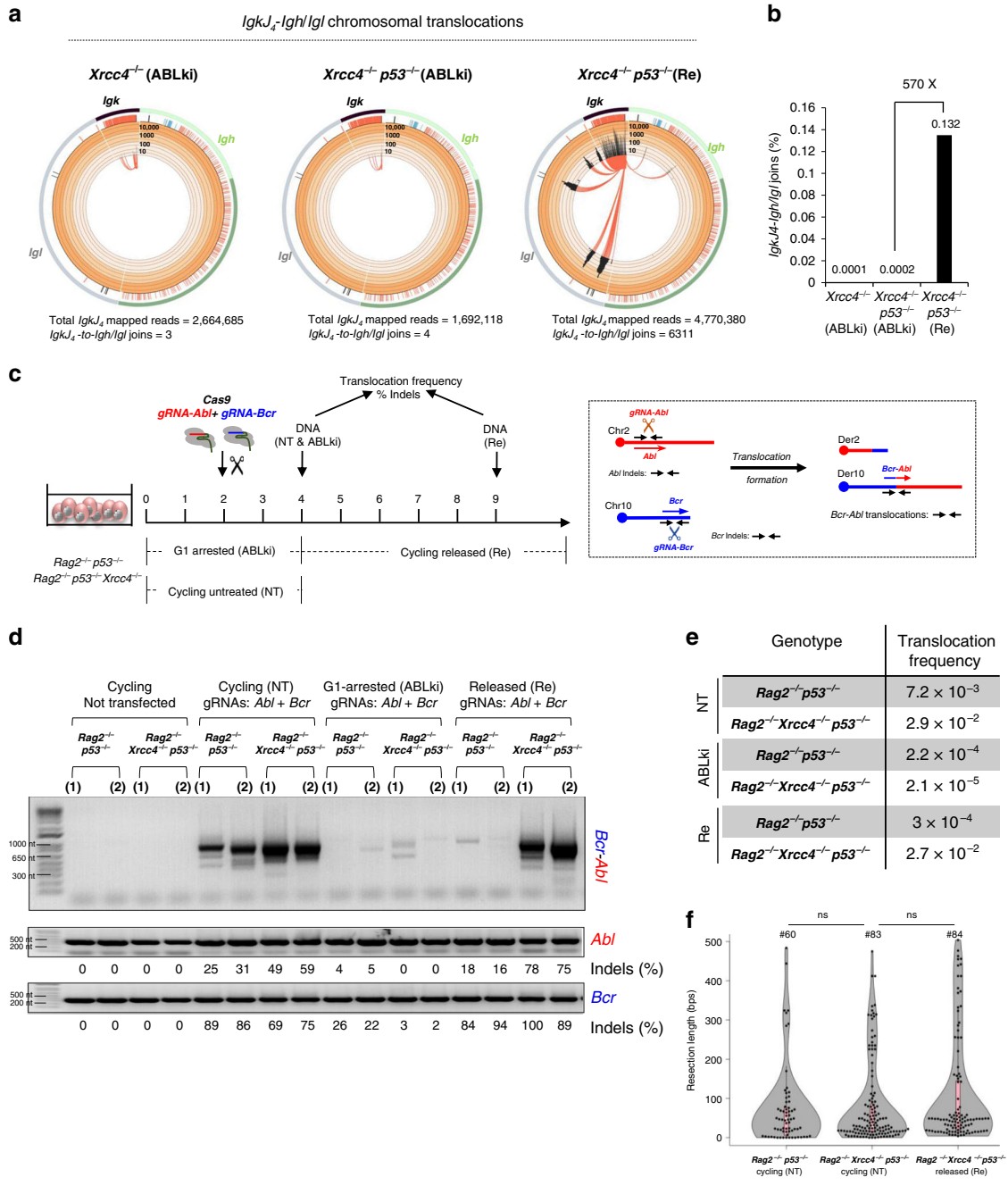

chromosomal translocations during cell cycle progression to S-G2/M.

**Alt-NHEJ of G1 DSBs in S-G2/M requires Pol θ but not PARP1.** We used CRISPR/Cas9-mediated gene editing to delete *Xrcc4* from previously reported *Parp1*[−/−] *Parp2*[f/f] *p53*[−/−] *v-Abl* pro-B cells (hereafter termed *Parp1*[−/−] *p53*[−/−] pro-B cells)[43], generating *Xrcc4*[−/−] *Parp1*[−/−] *p53*[−/−] pro-B cell clones. As expected[43], PARP activity was reduced in PARP1-deficient B cell clones as compared to PARP1-proficient B cell clones (Supplementary Fig. 11). Notably, *Xrcc4*[−/−] *Parp1*[−/−] *p53*[−/−] pro-B cell clones grew normally as compared to other cell types without noticeable changes in cell cycle or survival (Supplementary Fig. 12). To generate XRCC4/Pol θ/p53 deficient pro-B cells, we used CRISPR/Cas9-mediated gene editing to delete *Polq* (Δexon1)

from *p53*[−/−] pro-B cell lines, generating *Polq*[−/−] *p53*[−/−] pro-B cell clones. We subsequently deleted *Xrcc4* from *Polq*[−/−] *p53*[−/−] pro-B cells or *Polq* from the abovementioned *Xrcc4*[−/−] *p53*[−/−] pro-B cells, generating *Xrcc4*[−/−] *Polq*[−/−] *p53*[−/−] pro-B cell clones. Importantly, the isolated *Xrcc4*[−/−] *Polq*[−/−] *p53*[−/−] pro-B cell clones had a proliferative defect associated with a reduced fraction of cells in S phase and an increase in cell death that was counteracted by introduction of XRCC4, indicating that deficiencies in XRCC4 and Pol θ have synergistic deleterious effects on cell growth and viability (Supplementary Fig. 12).

We assessed the role of PARP1 and Pol θ in alt-NHEJ of unrepaired G1-DSBs by measuring *IgkV*$_{10-95}$*-J*$_4$ intrachromosomal rearrangements in released pro-B cells. *IgkV*$_{10-95}$*-J*$_4$ PCR amplification analysis revealed a substantial rescue of recombination in released *Xrcc4*[−/−] *Parp1*[−/−] *p53*[−/−] pro-B cells in the form of PCR amplified DNA products of variable sizes (Fig. 4a).

**Fig. 3 Unrepaired RAG- and Cas9-induced G1 DSBs promote translocations in S-G2-M. a** Circos plots displaying $IgkJ_4$-$Igl_{region}$/$Igh_{region}$ translocations distribution from $IgkJ_4$ coding end bait libraries. Junctions are represented as arcs originating from $IgkJ_4$ breaks with a minimum of 5 reads per 1,000 bp bin for $Xrcc4^{-/-}$ G1 arrested libraries, 4 reads per 1000 bp bin for $Xrcc4^{-/-}$ $p53^{-/-}$ G1 arrested libraries and 10 reads per 1000 bp bin for $Xrcc4^{-/-}$ $p53^{-/-}$ released libraries. Released cells were harvested 4 to 6 days after washing off ABLki. **b** Quantification of $IgkJ_4$-$Igl_{region}$/$Igh_{region}$ translocations in $Xrcc4^{-/-}$ and $Xrcc4^{-/-}$ $p53^{-/-}$ v-Abl pro-B cells. Values are the percentages of translocations relative to total mapped reads. Fold enrichment in $Xrcc4^{-/-}$ $p53^{-/-}$ released cells (Re) as compared to G1 blocked cells (ABLki) is indicated. Graph bars represent the pool of $n = 4$ independent experiments for $Xrcc4^{-/-}$ G1 blocked cells, $n = 3$ for $Xrcc4^{-/-}$ $p53^{-/-}$ G1 blocked cells and $n = 6$ for $Xrcc4^{-/-}$ $p53^{-/-}$ released cells with two independent cell lines for each genotype. See also Supplementary Table 4. **c** Left panel: schematic of DSB induction by Cas9 nuclease in cycling (NT) and G1-arrested (ABLki) v-Abl pro-B cells. Kinetics of ABLki treatment and release (Re) are indicated. Right panel: schematic of the PCR strategy used to amplify Abl and Bcr breakpoint junctions and Bcr-Abl chromosomal translocations. **d** Representative PCR amplifications of Bcr-Abl translocation breakpoints; in cycling (NT) (transfected or not with Cas9 and gRNAs), G1-arrested (ABLki) and released/cycling (Re) v-Abl pro-B cells of the indicated genotype. PCR of Abl and Bcr loci were used as controls and sequenced to evaluate InDels (from Supplementary Fig. 10D, E). Two independent cell lines were used for each genotype: $Rag2^{-/-}$ $p53^{-/-}$ (17585 & 17587); $Rag2^{-/-}$ $Xrcc4^{-/-}$ $p53^{-/-}$ (Xr-17585-17 & Xr-17585-18). The number of independent experiments is indicated in Supplementary Fig. 13C. Source data are provided as a Source Data file. **e** Translocation frequency of Bcr-Abl formation for the different genotypes in cycling (NT), G1-arrested (ABLki) and released (Re) conditions (from Supplementary Fig. 10B). **f** Violin plot representing resection lengths from Bcr and Abl breakpoint sites obtained from Bcr-Abl translocation junction sequences. Each dot indicates the resection length of a unique sequence obtained after TOPO cloning and sequencing with the total number (#) of sequences analyzed indicated (from Supplementary Table 5). No significance (ns) with $P > 0.05$ by Two-sided Student's t-test. $Rag2^{-/-}$ $p53^{-/-}$ cycling vs $Rag2^{-/-}$ $Xrcc4^{-/-}$ $p53^{-/-}$ cycling: $P = 0.702$; $Rag2^{-/-}$ $Xrcc4^{-/-}$ $p53^{-/-}$ cycling vs $Rag2^{-/-}$ $Xrcc4^{-/-}$ $p53^{-/-}$ released: $P = 0.085$.

By sharp contrast, $IgkV_{10-95}$-$J_4$ recombination products were absent from released $Xrcc4^{-/-}$ $Polq^{-/-}$ $p53^{-/-}$ pro-B cells (Fig. 4b). These results indicate that end joining of unrepaired G1 DSBs in released $Xrcc4^{-/-}$ $p53^{-/-}$ cells is independent of PARP1 and requires Pol θ.

We next measured RAG-induced genetic instability in released pro-B cells by performing Igk locus-specific DNA-FISH on chromosome spreads prepared from untreated cycling cells and G1-arrested cells released back into the cell cycle by removal of ABLki (Fig. 4c–e; Supplementary Table 6). Analysis of metaphases prepared from untreated cycling cells revealed very low levels of Igk-associated chromosome breaks and translocations in all genotypes (Supplementary Table 6). Released $Xrcc4^{-/-}$ $p53^{-/-}$ pro-B cells contained a statistically significant increase in aberrant metaphases (34.7%, $n = 813$) as compared with $p53^{-/-}$ (6.1%, $n = 745$) and $Xrcc4^{-/-}$ $p53^{-/-}$ pro-B cells complemented with wild type XRCC4 (7.4%, $n = 497$) (Fig. 4e; Supplementary Table 6). Specifically, over half of aberrant metaphases from released $Xrcc4^{-/-}$ $p53^{-/-}$ pro-B cells contained Igk locus-associated chromosomal translocations (18% of total metaphases) (Fig. 4d, e; Supplementary Table 6), which is in agreement with the high chromosomal translocation frequency detected by LAM-HTGTS (Fig. 3a, b). Similar to $Xrcc4^{-/-}$ $p53^{-/-}$ pro-B cells, analysis of metaphase spreads from released $Xrcc4^{-/-}$ $Parp1^{-/-}$ $p53^{-/-}$ pro-B cells revealed that 36.4% ($n = 257$) of the cells harbored Igk locus instability with approximately half being Igk locus-associated chromosomal translocations and the other half being chromosomal breaks (Fig. 4d, e; Supplementary Table 6). These results show that DNA end joining in the form of chromosomal translocations in released XRCC4/p53-deficient pro-B cells is independent of PARP1, supporting the conclusion that alt-NHEJ of unrepaired G1 DSBs in S-G2/M phase does not require PARP1.

In contrast to $Xrcc4^{-/-}$ $p53^{-/-}$ and $Xrcc4^{-/-}$ $Parp1^{-/-}$ $p53^{-/-}$ pro-B cells, released $Xrcc4^{-/-}$ $Polq^{-/-}$ $p53^{-/-}$ pro-B cells very rarely harbored chromosomal translocations (2.6%, $n = 373$) and, instead, contained Igk locus-associated chromosomal breaks at very high frequency (70.5%, $n = 373$) (Fig. 4d, e; Supplementary Table 6), demonstrating that Pol θ is required for alt-NHEJ of unrepaired RAG-induced G1 DSBs in S-G2/M. This was not due to a general DNA DSB repair defect in Pol θ-deficient cells as released $Xrcc4^{-/-}$ $Polq^{-/-}$ $p53^{-/-}$ pro-B cells complemented with XRCC4 and $Polq^{-/-}$ $p53^{-/-}$ pro-B cells harbored low levels of Igk instability (Fig. 4e; Supplementary Table 6). In addition, genetic instability at the Igk locus was nearly

absent from released (0.8%, $n = 792$) $Xrcc4^{-/-}$ $Polq^{-/-}$ $p53^{-/-}$ pro-B cells lacking RAG2 (Fig. 4e; Supplementary Table 6), indicating that the vast majority of unrepaired Igk breaks detected in released $Xrcc4^{-/-}$ $Polq^{-/-}$ $p53^{-/-}$ pro-B cells originated from G1 RAG-DSBs and not from another endogenous genotoxic source. Notably, during the course of our analysis, we noticed that fewer metaphases were obtained from released $Xrcc4^{-/-}$ $Polq^{-/-}$ $p53^{-/-}$ pro-B cells as compared with other cell types suggesting that unrepaired RAG-DSBs might lower the capacity of XRCC4/ Pol θ doubly deficient cells to cycle after ABLki removal (see below).

Importantly, as for RAG-induced chromosomal translocations, Cas9-induced Bcr-Abl chromosomal translocations in released XRCC4/p53-deficient pro-B cells were also dependent on Pol θ (chromosomal translocation frequency of ~$5.9 \times 10^{-4}$ in $Xrcc4^{-/-}$ $Polq^{-/-}$ $p53^{-/-}$ cells as compared to ~$2.7 \times 10^{-2}$ in $Xrcc4^{-/-}$ $p53^{-/-}$ cells) (Fig. 3d, e; Supplementary Figs. 10B and 13A–C). In addition, DNA-FISH analysis of metaphase spreads revealed high levels of Bcr-Abl chromosomal translocations in released in $Xrcc4^{-/-}$ $p53^{-/-}$ cells (20.4%, $n = 98$ metaphases) that were absent from the few metaphases we obtained from released $Xrcc4^{-/-}$ $Polq^{-/-}$ $p53^{-/-}$ cells (0/12 metaphases) (Supplementary Fig. 13D–F; Supplementary Table 6). Instead, released $Xrcc4^{-/-}$ $Polq^{-/-}$ $p53^{-/-}$ cells accumulated Cas9-induced Bcr or Abl chromosome breaks (8/12 metaphases) (Supplementary Fig. 13D–F; Supplementary Table 6). Together, these data demonstrate that Pol θ is essential for alt-NHEJ of unrepaired G1 DSBs, promoting chromosomal translocations and preventing the accumulation of broken chromosomes during cell cycle transition to M phase.

Interestingly, in contrast to released $Xrcc4^{-/-}$ $Polq^{-/-}$ $p53^{-/-}$ cells, Bcr-Abl chromosomal translocations were readily detected in cycling $Xrcc4^{-/-}$ $Polq^{-/-}$ $p53^{-/-}$ cells (Supplementary Fig. 13A–C), albeit with a translocation frequency that corresponds to approximately 50% of the one observed in $Xrcc4^{-/-}$ $p53^{-/-}$ cells (~$1.5 \times 10^{-2}$ and ~$2.9 \times 10^{-2}$, respectively) (Fig. 3d, e; Supplementary Figs. 10B and 13A–C), suggesting that, in XRCC4/p53-deficient cycling cells, Pol θ is only responsible for a fraction of the chromosomal translocations originating at Cas9-DSBs. DNA-FISH analysis on metaphase spreads confirmed this tendency with 6.8% of metaphases ($n = 143$) containing Bcr-Abl translocations in Cas9-transfected $Xrcc4^{-/-}$ $Polq^{-/-}$ $p53^{-/-}$ cells as compared to 16.9% ($n = 188$) in $Xrcc4^{-/-}$ $p53^{-/-}$ cells (Supplementary Fig. 13D–F; Supplementary Table 6). Most notably, this decrease in Bcr-Abl translocations was accompanied

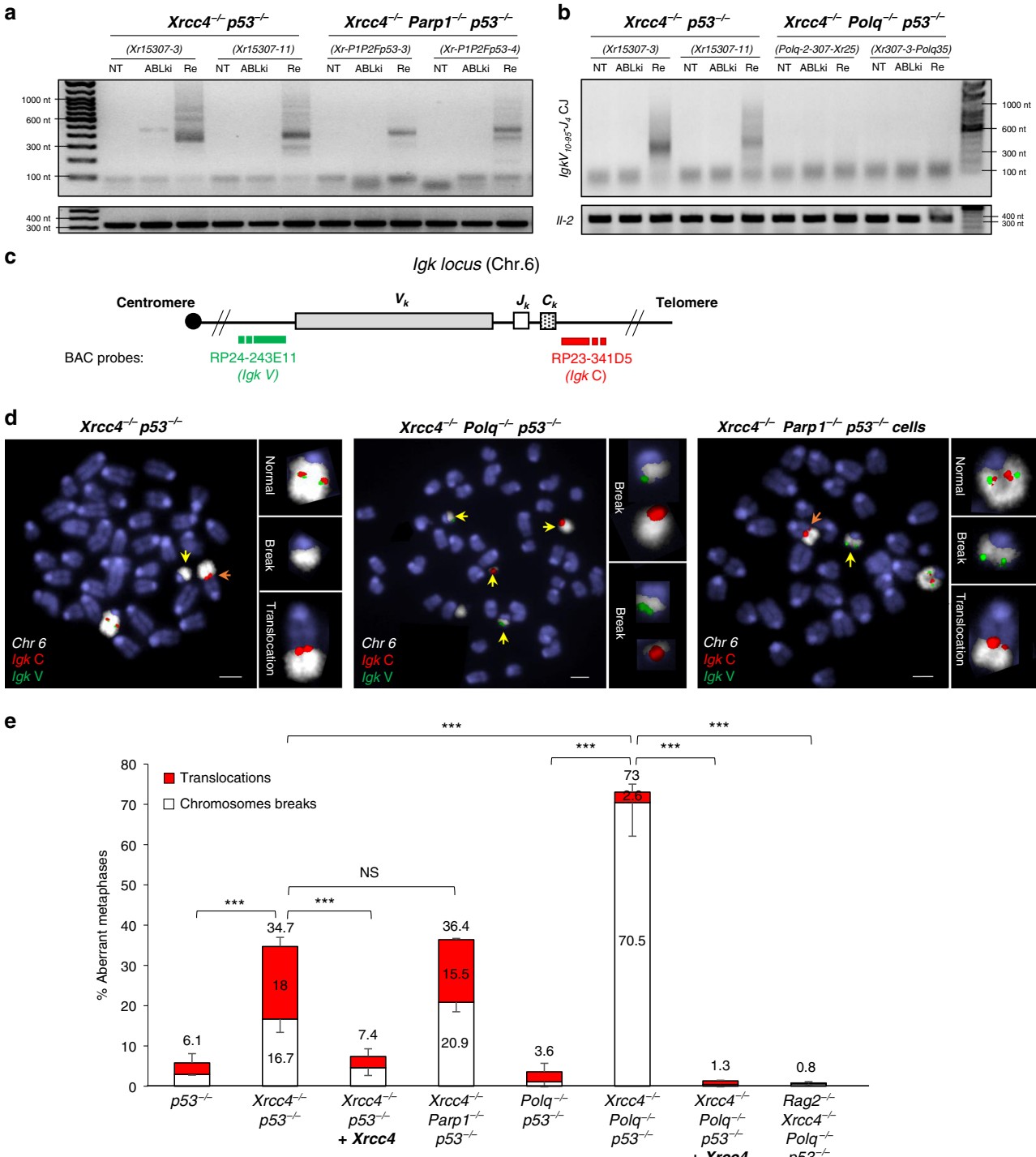

**Fig. 4 Pol θ promotes alt-NHEJ of unrepaired G1-DSBs and translocations in S-G2/M phase. a**, **b** Representative semi-quantitative nested PCR analysis of *Igk* $V_{10-95}$-*to*-$J_4$ coding joins from untreated (NT), G1-blocked (ABLki) and released (Re) *v-Abl* pro-B cells of the indicated genotype. *Il-2* gene PCR was used as a loading control. Released cells were harvested 5 days after washing off ABLki. Two independent cell lines were used for each genotype: *Xrcc4⁻/⁻ p53⁻/⁻* (Xr15307-3 & Xr15307-11); *Xrcc4⁻/⁻ Parp1⁻/⁻ p53⁻/⁻* (Xr-P1P2Fp53-3 & Xr-P1P2Fp53-4); *Xrcc4⁻/⁻ Polq⁻/⁻ p53⁻/⁻* (Polq-2-307-Xr25 & Xr307-3-Polq35). Source data are provided as a Source Data file. The PCR gels are representative of *n* = 2 independent experiments. **c** Schematic representation of the *Igk* locus, with positions of the BACs used for generation of DNA FISH probes indicated. **d** Representative metaphases from released *Xrcc4⁻/⁻ p53⁻/⁻*, *Xrcc4⁻/⁻ Polq⁻/⁻ p53⁻/⁻* and *Xrcc4⁻/⁻ Parp1⁻/⁻ p53⁻/⁻* *v-Abl* pro-B cells with *Igk* C BAC probe (red), *Igk* V BAC probe (green) and chromosome 6 paint (white). Yellow and orange arrowheads point to broken and translocated chromosome 6, respectively. Scale bars = 3 μm. The number of analyzed metaphases for each condition is indicated in Supplementary Table 6. **e** Percentage of aberrant metaphases from released *v-Abl* pro-B cells of the indicated genotypes. Released cells were harvested 5 days after washing off ABLki. Histograms represent means ± s.d. from at least two independent experiments performed with a minimum of two independent B cell clones (the list of independent experiments and total metaphases analyzed are provided in Supplementary Table 6). The mean percentage of total aberrant metaphases is indicated above. The mean percentages of metaphases with translocations (red) and chromosomes breaks (white) are indicated (See also Supplementary Table 6). ***$P$ < 0.001 (Two-sided Fisher exact test).

with an increase in the percentage of metaphases carrying *Bcr* or *Abl* chromosome breaks in *Xrcc4*$^{-/-}$ *Polq*$^{-/-}$ *p53*$^{-/-}$ cells as compared to *Xrcc4*$^{-/-}$ *p53*$^{-/-}$ cells (31.8% and 13.5%, respectively) (Supplementary Fig. 13D–F; Supplementary Table 6). These results are in agreement with the conclusion that unrepaired G1-induced DSBs require Pol θ to promote chromosomal translocations and to suppress the accumulation of broken chromosomes in S-G2/M. In addition, these data indicate that in cycling XRCC4/p53-deficient cells a fraction of the Cas9-DSBs generate chromosomal translocations independently of Pol θ with these DSBs being generated in S-phase cells.

**Synthetic lethality between Pol θ and XRCC4 after G1 DSBs**. To examine the regulation of Pol θ during the cell cycle, we next quantified its expression by RT-PCR in pro-B cell clones. Interestingly, although *Polq* expression was readily detected in unsynchronized cycling pro-B cells, it was undetectable from G1-arrested pro-B cells. Importantly, *Polq* expression increased as cells re-entered S phase indicating that its expression is repressed in G1-arrested cells and induced during G1/S cell cycle entry (Fig. 5a, b; Supplementary Fig. 14).

We next measured survival of *Xrcc4*$^{-/-}$ *Polq*$^{-/-}$ *p53*$^{-/-}$ pro-B cells and control pro-B cell clones following ABLki wash off. Strikingly, as opposed to *p53*$^{-/-}$, *Polq*$^{-/-}$ *p53*$^{-/-}$, *Xrcc4*$^{-/-}$ *p53*$^{-/-}$ and *Xrcc4*$^{-/-}$ *Polq*$^{-/-}$ *p53*$^{-/-}$ complemented with XRCC4 pro-B cells that grew exponentially 3 to 4 days after ABLki removal and displayed minimal cell death (approximately 5 to 10%), *Xrcc4*$^{-/-}$ *Polq*$^{-/-}$ *p53*$^{-/-}$ pro-B cells underwent a maximum of one cell division and suffered dramatic increase in cell death (42%) 5 days after ABLki removal (Fig. 5c, d). As we observed an accumulation of *Igk* locus-associated chromosomal breaks in metaphases obtained from released *Xrcc4*$^{-/-}$ *Polq*$^{-/-}$ *p53*$^{-/-}$ pro-B cells (Fig. 4d, e), we hypothesized that unrepaired RAG-induced DSBs might cause the observed loss of cell viability during cell division. To test this, we measured the viability of *Xrcc4*$^{-/-}$ *Polq*$^{-/-}$ *p53*$^{-/-}$ pro-B cells lacking RAG2. In contrast to RAG-proficient *Xrcc4*$^{-/-}$ *Polq*$^{-/-}$ *p53*$^{-/-}$ pro-B cells, *Rag2*$^{-/-}$ *Xrcc4*$^{-/-}$ *Polq*$^{-/-}$ *p53*$^{-/-}$ pro-B cells grew exponentially 4 to 6 days after ABLki removal with significantly lower cell death (approximately 13%) (Fig. 5c, d). Together, these results indicate that RAG-induced DSBs generated in G1-arrested pro-B cells cause the observed synthetic lethality between XRCC4 and Pol θ during cell cycle entry. To further address this issue, we performed a reciprocal experiment by complementing *Rag2*$^{-/-}$ *Xrcc4*$^{-/-}$ *Polq*$^{-/-}$ *p53*$^{-/-}$ pro-B cells with a retroviral vector encoding for RAG2 and the cell surface marker Thy1.1. We mixed complemented Thy1.1$^+$ pro-B cells and un-complemented Thy1.1$^-$ pro-B cells at a ratio of 1:1 and followed the percentage of Thy1.1-positive and Thy1.1-negative cells during normal cell culture conditions and after release from ABLki treatment (Supplementary Fig. 15A, B). Strikingly, as opposed to control cell culture conditions in which Thy1$^+$ and Thy1.1$^-$ cells remained at a 1-to-1 ratio, Thy1.1$^+$ RAG2-expressing *Xrcc4*$^{-/-}$ *Polq*$^{-/-}$ *p53*$^{-/-}$ pro-B cells specifically and gradually disappeared from the cell populations after ABLki treatment (Supplementary Fig. 15B). Deficiencies in XRCC4 and Pol θ thus have dramatic synergistic effects on cell growth and viability upon induction of RAG-DSBs in G1 cells, the doubly deficient cells being synthetic dead.

As *v-Abl* transformed pro-B cells serve as correlates to BCR-ABL-positive B-cell acute lymphoblastic leukemia[44], we next tested whether RAG-induced DSBs could sensitize XRCC4/Pol θ doubly deficient leukemic cells to the ABL kinase inhibitor Dasatinib in vivo. To test this hypothesis, we injected intravenously 0.5 × 10$^6$ leukemic pro-B cells per mouse that included

either *Xrcc4*$^{-/-}$ *p53*$^{-/-}$ pro-B cells ($n = 24$ mice) or *Xrcc4*$^{-/-}$ *Polq*$^{-/-}$ *p53*$^{-/-}$ pro-B cells ($n = 17$ mice). After 7 days of incubation, treatment with Dasatinib (10 mg/kg) or with solvent only (mock) was initiated once per day by gavage for a period of 21 days (Fig. 5e). In mock-treated conditions, the animals showed first signs of disease, such as reduced mobility and scrubby fur, within 17 days post-injection for mice injected with *Xrcc4*$^{-/-}$ *p53*$^{-/-}$ leukemic B cells as compared to 27 days for mice injected with *Xrcc4*$^{-/-}$ *Polq*$^{-/-}$ *p53*$^{-/-}$ leukemic B cells (Fig. 5f), which is consistent with the slower cell growth observed for *Xrcc4*$^{-/-}$ *Polq*$^{-/-}$ *p53*$^{-/-}$ pro-B cells in vitro (Supplementary Fig. 12A). Analysis of lymphoid organs from sick animals revealed the presence of proliferating CD19$^+$ CD43$^+$ lymphoblastic cells for both *Xrcc4*$^{-/-}$ *p53*$^{-/-}$ and *Xrcc4*$^{-/-}$ *Polq*$^{-/-}$ *p53*$^{-/-}$ leukemic B cells (Supplementary Fig. 16). Strikingly, Dasatinib treatment of mice injected with *Xrcc4*$^{-/-}$ *p53*$^{-/-}$ leukemic B cells only partially extended survival with 50% of the animals succumbing 37 days post-injection which corresponds to 9 days after the end of the treatment (Fig. 5f). In sharp contrast, Dasatinib treatment showed a much better response in animals injected with *Xrcc4*$^{-/-}$ *Polq*$^{-/-}$ *p53*$^{-/-}$ leukemic B cells with 50% of the mice surviving 88 days post-injection which corresponds to 60 days after the treatment was stopped. Importantly, 4 out of 9 mice survived without signs of the disease 100 days after injection, indicating that the vast majority of the leukemic cells were killed within the course of the treatment (Fig. 5f). Together, these data demonstrate that induction of G1 DNA damage by the ABL kinase inhibitor Dasatinib eradicates leukemic cells that are deficient for canonical and pol θ-dependent alternative NHEJ.

## Discussion

While two canonical DSB repair mechanisms—NHEJ and HR—have been shown to operate and antagonize one another during the cell cycle, alternative repair pathways exist and take place in cells deficient for one or multiple components of the NHEJ or HR machineries. Of particular interest is the alt-NHEJ pathway that has been implicated in the generation of aberrant chromosomal rearrangements and shown to promote the survival of NHEJ- or HR-deficient cells. Given the increasing interest in pharmacologically targeting alt-NHEJ in certain cancers, its mode of action during different phases of the cell cycle is an important open question.

In contrast to other types of DSBs which, in most cases, are amenable to repair by multiple pathways, RAG-DSBs are joined almost exclusively by NHEJ. This restriction is thought to rely on at least two distinct mechanisms; (1) the ability of RAG to channel broken DNA ends towards the NHEJ pathway and, (2) the periodic destruction of RAG2 at the G1-to-S transition thus restricting DSB repair to the G1 phase of the cell cycle when DNA ends access to homology-based repair pathways is limited[2,30,45]. Using high-throughput sequencing assays, we find that end joining of RAG-induced DSBs in G1-arrested XRCC4-deficient pro-B cells is virtually null. Our observation that Cas9-induced DSBs are also poorly repaired and do not generate chromosomal translocations in these cells support the conclusion that restriction factors other than RAG limit access of broken DNA ends to alt-NHEJ in G1-phase cells. Ku, H2AX, 53BP1 and possibly downstream Shieldin factors might act to limit resection in G1 cells, a requirement for alt-NHEJ[24,46,47]. In addition, as Pol θ expression is limited to S-G2/M, absence of Pol θ in G1 cells might also prevent efficient alt-NHEJ in G1-arrested XRCC4-deficient cells.

Regardless of the underlying restriction mechanisms that act in G1 cells, we show that the progression of unrepaired G1-induced DSBs into S-G2/M restores the cellular DSB repair capacities. The

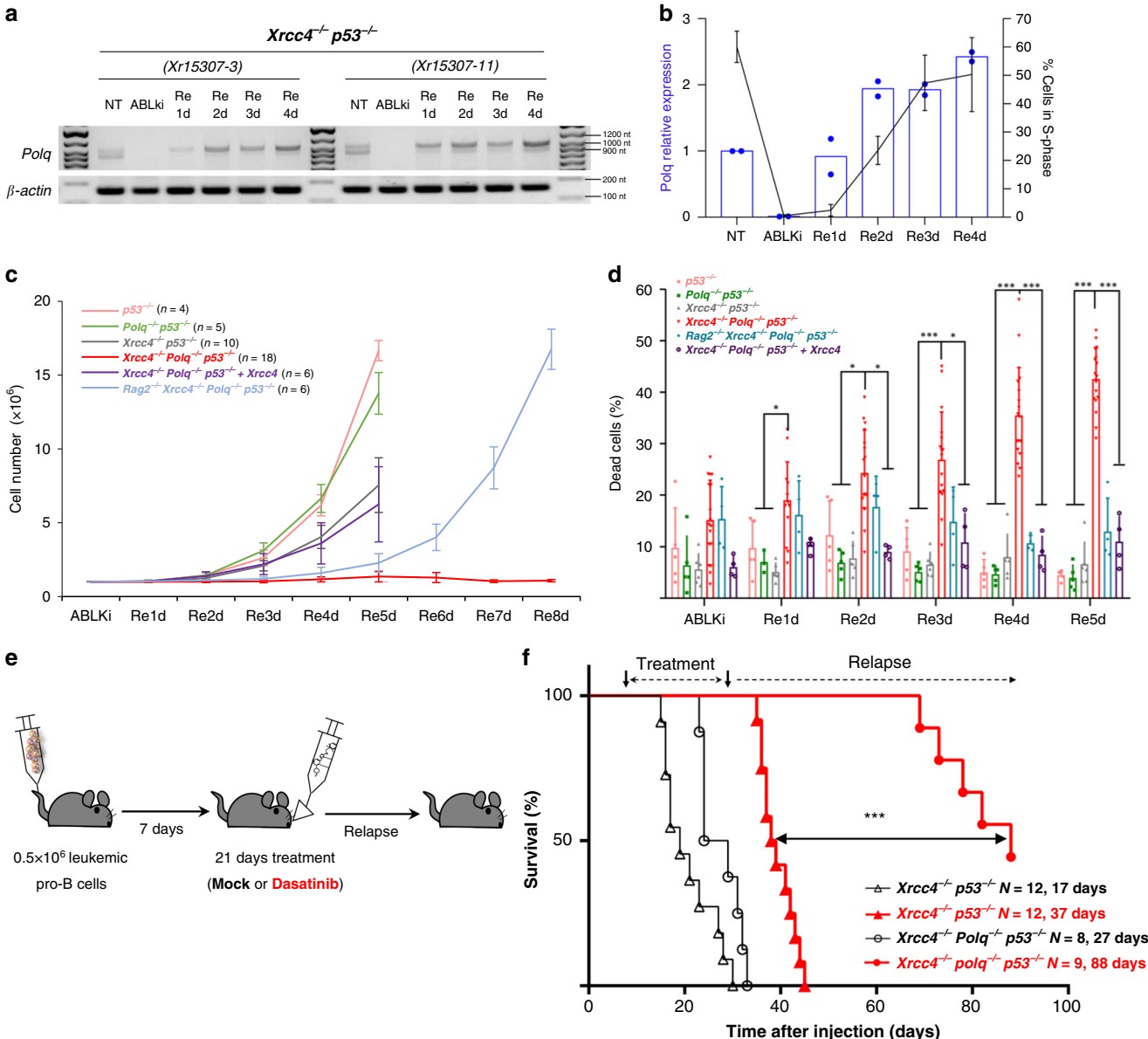

**Fig. 5 Pol θ-mediated alt-NHEJ promotes the survival of NHEJ-deficient leukemic B cells after G1 DSBs. a** *Polq* mRNA expression in untreated (NT), G1-blocked (ABLki) and released (Re1d to Re4d) *Xrcc4$^{-/-}$ p53$^{-/-}$* v-Abl pro-B cells. β-*actin* mRNA was used as a control. Cell lines used: *Xrcc4$^{-/-}$ p53$^{-/-}$* (*Xr15307-3 & Xr15307-11*). **b** Graph displaying the correlation between *Polq* mRNA expression (blue bar graph, left y axis) and the percentage of S-phase cells (black curve graph, right *y*-axis) in *Xrcc4$^{-/-}$ p53$^{-/-}$* v-Abl pro-B cells. NT: untreated; ABLki: treated for 72 h with ABLki; Re1d to Re4d: released. *Polq* mRNA expression data represent the means of two independent *Xrcc4$^{-/-}$ p53$^{-/-}$* v-Abl pro-B cell clones (*Xr15307-3* and *Xr15307-11*) and are normalized to β-*actin*. **c** Cell counts for G1 blocked cells (ABLki) and released cells (Re - 1 to 8 days after released from ABLki treatment). Data represent the mean ± SEM. The number of independent experiments for each genotype is indicated into brackets. **d** Percentage of dead cells in G1 blocked (ABLKi) and released cells (Re - 1 to 5 days after released from ABLki treatment). Data represent the mean + SEM. *P < 0.05, ***P < 0.001 (Two-sided Student's t-test). The number of independent experiments for each condition is indicated in the source data file. **e** 0.5 million *Xrcc4$^{-/-}$ p53$^{-/-}$* v-Abl pro-B cells or *Xrcc4$^{-/-}$ Polq$^{-/-}$ p53$^{-/-}$* v-Abl pro-B cells were injected in *Rag2$^{-/-}$* mice. 7 days after injection, mice were treated with Dasatinib or with solvent (mock) for 21 consecutive days. **f** Kaplan–Meier survival analysis for cohorts of *Rag2$^{-/-}$* animals injected with *Xrcc4$^{-/-}$ p53$^{-/-}$* v-Abl pro-B cells (*Xrcc4$^{-/-}$ p53$^{-/-}$*, n = 24) or *Xrcc4$^{-/-}$ Polq$^{-/-}$ p53$^{-/-}$* v-Abl pro-B cells (*Xrcc4$^{-/-}$ Polq$^{-/-}$ p53$^{-/-}$*, n = 17) and that were either mock-treated (black lines) or treated with Dasatinib at 10 mg/kg (red lines). Animals were monitored for 100 days. The median age of sacrifice (50% survival) after injection is shown for *Xrcc4$^{-/-}$ p53$^{-/-}$* (mock, 17 days; Dasatinib, 37 days) and *Xrcc4$^{-/-}$ Polq$^{-/-}$ p53$^{-/-}$* (mock, 27 days; Dasatinib, 88 days) genotypes. ***P = 0.000004 (Two-sided Log-rank test). Source data are provided as a Source Data file.

progression of unrepaired breaks from G1- to S-phase requires inactivation of the p53-dependent G1/S checkpoint that enables DNA ends to access error-prone alt-NHEJ. This is illustrated by the presence of repair products (i.e., *Igk* rearrangements) bearing kilobase long DNA end resection and micro-homologies in released XRCC4/p53-deficient pro-B cells. This hyper-resection is consistent

with the idea that DNA end protection mechanisms that are in place in G1 cells no longer act as a barrier to alternative end joining pathways when the cell enters S phase[2]. Additionally, G1-induced broken DNA ends generate chromosomal translocations at high frequency in S-G2/M, indicating that synapsis between the broken DNA ends is also lost during cell cycle progression.

Analysis carried out in this study are consistent with previous work showing that chromosomal translocation formation in mouse cells primarily arises by alt-NHEJ[42,48]. First, G1-arrested and released NHEJ-proficient cells harbored very low levels of chromosomal translocations after RAG- or Cas9-induced DSBs. Second, when assessed in unsynchronized cycling cells, Cas9-induced chromosomal translocation frequency increased approximately 4-fold in XRCC4-deficient cells as compared to XRCC4-proficient cells. Third, in cycling conditions, analysis of Cas9-induced translocation breakpoint junctions revealed similar patterns of resection and microhomology in XRCC4-proficient and XRCC4-deficient cells indicating that alt-NHEJ is the main driver of chromosomal translocations in both cell types.

In contrast, we have previously shown that breakpoint junctions for a cancer chromosomal translocation induced by specific engineered nucleases in NHEJ-proficient human cells are characterized by little or no processing. Moreover, in these settings, the loss of LIG4 does not increase translocation frequency suggesting that canonical NHEJ is primarily implicated in these events[5]. Nevertheless, alt-NHEJ is operative in human cells and becomes critical to form chromosomal translocations in the absence of LIG4 or XRCC4 as shown by the nature of the breakpoint junctions[5]. Human tumors can be generally classified either as containing few chromosomal aberrations, for instance one specific chromosomal translocation, or as containing highly rearranged genomes in the form of complex chromosomal rearrangements. We speculate that initial oncogenic rearrangements such as balanced chromosomal translocations may arise from canonical NHEJ, possibly in G1-arrested cells or slow proliferating cells. Because more advanced tumor cells are generally highly proliferative and often harbor deficiencies in DNA damage response and repair pathways, based on our results presented in this study, we suggest that the accumulation of chromosomal rearrangements in these cells might be fueled by alt-NHEJ in S-G2/M and that these events would give rise to tumors containing complex karyotypes. In support of this, loss of LIG4 or XRCC4 in the context of p53-deficiency was recently reported to be associated with a more frequent occurrence of complex genome rearrangements in glioblastoma[49].

The relative contribution of Pol θ and PARP1 to the formation of chromosomal translocations and whether they work together in alt-NHEJ remains an active topic of discussion[25,31,32,50,51].

In naïve splenic B cells stimulated for immunoglobulin class-switching, Pol θ-deficiency leads to an increase in *Igh/Myc* chromosomal translocation frequency that is not observed in stimulated B cells from PARP1 knockout mice[50,51]. Pol θ can also suppress translocations between a pair of Cas9-induced breaks targeted to different chromosomes in transformed mouse embryonic fibroblasts, although a significant effect was limited to cells already deficient for NHEJ[31]. In contrast, another study assessing Cas9-induced translocations in mouse pluripotent stem cells showed that these were promoted by Pol θ and, similarly, that Pol θ was required in NHEJ-deficient mouse embryonic fibroblasts for fusion of chromosomes at de-protected telomeres. In this study, it was also reported that PARP1 facilitates the recruitment of Pol θ to DSBs to promote alt-NHEJ, implying that they are in the same pathway[32]. These studies indicate that Pol θ both suppresses and promotes DSB-induced chromosomal translocations and that this discrepancy may reflect differences in the origin of the chromosome break, cell type and possibly other biological parameters.

Our results highlight the importance of the cell cycle in the regulation of Pol θ-dependent and Pol θ-independent alt-NHEJ. We find that in the absence of XRCC4 and p53, DSBs generated in G1-arrested cells require cell cycle entry into S-phase for error-prone repair by alt-NHEJ in a manner that is dependent on Pol θ.

In contrast, DSBs generated in unsynchronized cells promote chromosomal translocations through both Pol θ-dependent and Pol θ-independent alt-NHEJ, indicating that, as opposed to G1-DSBs progressing to S-phase, DSBs generated in S-phase cells are repaired independently of Pol θ. These results are consistent with a recent study showing that in the absence of both LIG4 and Pol θ, Cas9-editing is still possible albeit with reduced efficiency (10 to 20%)[52]. These findings also indicate that an additional DNA repair mechanism, possibly single-strand annealing or another homology-directed pathway, is capable of joining DSBs generated in S-phase. Such pathway is likely to be responsible for the survival of NHEJ/Pol θ-doubly deficient cells in normal cell growth conditions.

In contrast to Pol θ, we find that alt-NHEJ of G1-DSBs in S-G2/M does not require PARP1 as illustrated by the presence of RAG-induced *Igk* cis-rearrangements and translocations in XRCC4/PARP1/p53-deficient cells released into the cell cycle. Thus, with regard to this particular type of DNA damage, Pol θ performs end joining activities that are independent of PARP1. PARP1 has been proposed to promote alt-NHEJ by competing with the Ku complex for access to DNA ends[29,53–56] and thus might not be operative in XRCC4-deficient cells. Interestingly, in the absence of XRCC4, alt-NHEJ is virtually null—or present at extremely low levels—in G1-arrested pro-B cells. Whether Ku blocks access of DNA ends to alternative end joining pathways and possibly PARP1-dependent alt-NHEJ in XRCC4-deficient G1 cells remains to be determined. In addition, although we show that PARP1 is dispensable for the joining of unrepaired G1-DSBs progressing to S-phase, it will be interesting to test whether the trapping of PARP on DNA by PARP inhibitors interferes with Pol θ-dependent alt-NHEJ.

Notably, we show that a combined deficiency in XRCC4 and Pol θ leads to synthetic lethality in response to G1-induced DSBs. Such induced lethality is not observed in XRCC4 single knockout cells or XRCC4/PARP1 doubly deficient cells, leading us to hypothesize that Pol θ might represent a promising target for killing NHEJ-deficient cancer cells exposed to G1 DNA damage. In support of this, we find that triggering endogenous RAG-induced DNA damage by mean of the Abelson tyrosine-kinase inhibitor Dasatinib selectively eliminates XRCC4/Pol θ /p53-deficient leukemic B cells in a mouse model of BCR-ABL leukemia. As certain cancers display dysfunctional NHEJ that correlates with increased genetic instability and aggressiveness[49,57–59], our findings point toward therapeutic opportunities to target Pol θ in combination with DSB-inducing agents in such tumors (Fig. 6).

## Methods

**Mice**. *Rag2*[−/−] mice (Taconic) were used for tumor cell injections. *Rag2*[−/−] mice (Taconic) were bred with *p53*[+/−] mice (Jackson laboratory[60]) to generate doubly deficient mice used to produce *v-Abl* transformed *Rag2*[−/−] *p53*[−/−] pro-B cell lines[7]. Mice were bred and housed at ambient temperature and humidity with 12 h light/12 h dark cycles. All experiments were performed in accordance with the guidelines of the institutional animal care and ethical committee of Institut Pasteur/CETEA n°89 under the protocol numbers 180006/14778 and 170027.

**Generation and culture of *v-Abl* pro-B cell lines**. *v-Abl* pro-B cell lines were generated as previously described[35,61]. Briefly, total bone marrow from 3–5-week-old mice was cultured and infected with a retrovirus encoding *v-Abl* kinase to generate immortalized pro-B cell lines[62]. *v-Abl* transformed pro-B cell lines were then transduced with pMSCV-Bcl2-puro retrovirus[63] to protect them from *v-Abl* kinase inhibitor-induced cell death. *v-Abl* pro-B cells were maintain in RPMI 1640 supplemented with 10% fetal bovine serum (Sigma F6178), penicillin (100 U/ml)/streptomycin (100 μg/ml) and 50 μM 2-mercaptoethanol (Gibco 11528926).

**CRISPR-Cas9 editing of *v-Abl* pro-B cells**. *v-Abl* pro-B knockout cell clones were generated as previously described[61,64]. Briefly, cells were nucleofected with two sgRNA-encoding plasmids and the pCas9-GFP plasmid (see Supplementary Table 2 for sgRNAs sequences). Electroporated cells were left to recover for 36-48 h

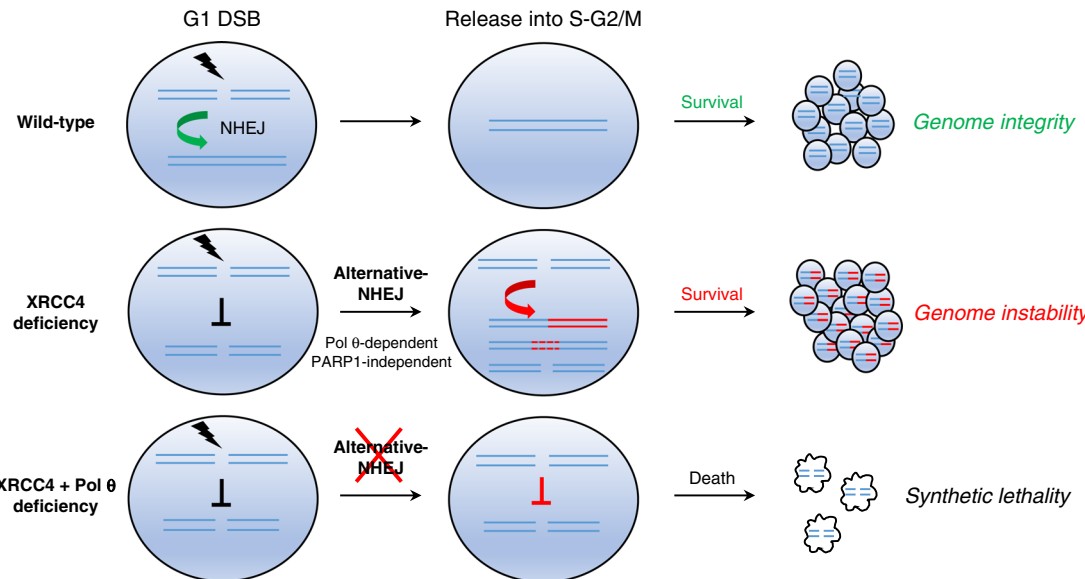

**Fig. 6 Model of DSB repair and synthetic lethality in ABL transformed p53-deficient pro-B cells exposed to G1 DNA damage.** Top panel: XRCC4 proficient (wild type) cells repair DSBs in G1 and maintain genome integrity upon release into the cell cycle. Middle panel: XRCC4-deficient cells are unable to repair DSBs in G1. In the absence of p53 G1/S checkpoint, DSBs are repaired in S-G2/M by Pol θ-dependent and PARP1-independent alternative NHEJ that enables cell survival and promotes genomic instability in the form of long DNA resection and chromosomal translocations. Bottom panel: In the absence of XRCC4- and Pol θ-dependent repair, G1 DSBs cause p53-independent cell death due to the accumulation of unresolved DNA breaks in S-G2-M.

and single cells expressing GFP were sorted in 96-well plates. Clones were then screened by PCR (See Supplementary Table 2 for primers sequences) and purified PCR products were Sanger sequenced to identify CRISPR/Cas9-induced deletions and insertions at the cleavage site. A minimum of two independent knock-out clones were generated (See Supplementary Table 1).

**V(D)J recombination assay.** V(D)J recombination assays were performed as previously described[35,61]. Briefly, v-Abl pro-B cells were transduced with pMX-INV retroviral vector and cells that had integrated the pMX-INV recombination substrate were enriched based on hCD4 expression using hCD4 Microbeads (Miltenyi 130-045-101). Purified pMX-INV v-Abl pro-B cells were treated with 3 μM of the Abl kinase inhibitor STI-571 (Novartis) for 72 h and assayed for recombination levels by FACS analysis of GFP/hCD4 expression (hCD4-PE antibody, Miltenyi 130-113-254, clone M-T466, 1:100). V(D)J recombination levels were scored by FACS analysis as the percentage of GFP positive cells among total hCD4 positive cells. FACS data was acquired by FACS Diva v8.0.1 (BD Biosciences) and analyzed and visualized by Flowjo v10.4.2 (TreeStar).

**PCR analysis of V(D)J recombination products.** Endogenous $Vk_{10-95}/Jk_4$ coding joins (CJs) were amplified by nested-PCR as previously described for $Vk_{6-23}/Jk_1$[35]. A total of 500 ng of genomic DNA was amplified using JK4-pa and V10-95-pa primers and the following cycles: 1× (95 °C 5 min); 17× (94 °C 3 s, 62 °C 30 s, 72 °C 30 s); 1× (72 °C 5 min). Serial 4-fold dilutions of this reaction were amplified using JK4-pb and V10-95-pb primers and the following cycles: 1× (95 °C 5 min); 30× (94 °C 30 s, 62 °C 30 s, 72 °C 30 s); 1× (72 °C 5 min). Il2 gene was amplified using IMR42 and IMR43 primers and was used as loading control. The PCR gel images were acquired and analyzed using Image Lab v6.0 (Biorad). See Supplementary Table 2 for primers sequences.

**TOPO cloning of V(D)J recombination products.** TOPO cloning was performed using TOPO® TA Cloning® kit (Invitrogen 450640) following manufacturer's instructions and analyzed by Sanger Sequencing using T7 and SP6 primers (see Supplementary Table 2).

**Cell cycle assay.** Cells were incubated with 100 μM Edu (Jena Bioscience CLK-N001-25) for 30 min at 37 °C and washed once with PBS. Cells were then fix cells with 4% Paraformaldehyde (Thermo Fisher Scientific AA433689M) for 30 min at room temperature and washed once with PBS. Cells were stained with 1 mM CuSO4 (Sigma C1297-100G), 2 μM iFluor™ 647 azide (AAT Bioquest 1091) and 100 mM L-Ascorbic acid (Sigma A5960-25G) for 1 h at room temperature. After one wash with PBS, cells were incubated with PI staining cocktail (1.9 mM sodium citrate tribasic dehydrate (Sigma S4641-500G), 25 μg/ml propidium iodide (ThermoFisher Scientific 440300250), 250 μg/ml RNAse A (Invitrogen 8003089), 0.5 mM Tris-Hcl, 0.75 mM NaCl) for 1 h at 37 °C. Finally, cells were centrifuged

and resuspended in PBS for FACS analysis. FACS data was acquired by FACS Diva v8.0.1 (BD Biosciences) and analyzed and visualized by Flowjo v10.4.2 (TreeStar).

**PARP activity.** Protein extracts were prepared from v-Abl pro-B cells as previously described[65]. PARP activity was determined by using 10 μg of total protein and the HT Universal Colorimetric PARP Assay Kit with Histone-Coated Strip Wells (Trevigen) following manufacturer's instructions.

**DNA FISH on metaphase spreads.** DNA FISH was performed as previously described[61]. Slides were treated with RNase A for 40 min, dehydrated in 70, 90 and 100% ethanol for 3 min each, denatured in 70% formamide/2× SSC for 3 min at 77 °C, dehydrated again in cold ethanol series, and hybridized with probes o/n at 37 °C in a humid chamber. The next day, slides were washed three times in 50% formamide/0.5X SSC for 5 min each at 37 °C and twice in 0.5× SSC for 10 min each at 37 °C. Finally, slides were mounted in ProLong Gold antifade reagent with DAPI (Invitrogen P36931) to counterstain total DNA. Metaphases were imaged using a ZEISS AxioImager.Z2 microscope and the Metafer automated capture system (MetaSystems), and counted manually. Probes used in this study: Igk C BAC probe (RP23-341D5), Igk V BAC probe (RP24-243E11), Abl BAC probe (RP23-65P13) and Bcr BAC probe (RP23-115L17).

**RT–PCR.** Total RNA was isolated from $3 \times 10^6$ v-Abl pro-B cells using the RNeasy Mini Kit (Qiagen 74104) according manufacturer's instructions and cDNA was synthesized using the High-Capacity cDNA Reverse Transcription Kit (Thermo-Fisher Scientific 4368814). Polq cDNA was amplified by semi-quantitative RT–PCR using Polq-F and Polq-R primers[66] (see Supplementary Table 2) and the following cycles: 1× (95 °C 5 min); 30× (95 °C 30 s, 60 °C 30 s, 72 °C 90 s); 1× (72 °C 5 min). β-actin was used as a control for transcript expression. The PCR gel images were acquired and analyzed using Image Lab v6.0 (Biorad).

**v-Abl pro-B cells transplantation and Dasatinib treatment.** Cells were cultured for 5 days and assessed for viability (at least 95% of viable cells) before transplantation. Cells were washed once in PBS and resuspended in PBS at 2.5 millions/ml before injection (200 μl, $0.5 \times 10^6$ cells) in the tail vein of 7–8-week-old $Rag2^{-/-}$ mice. 100 mM Dasatinib stock solution was prepared in DMSO and stored at −20 °C. 2 mM working Dasatinib solution was prepared in 80 mM citric acid (pH 3.1) and used immediately. Mice were fed with 10 mg/kg Dasatinib or an equivalent volume of DMSO /citric acid solution by oral gavage once per day for 21 consecutive days starting at day 7 post-injection.

**Complementation of v-Abl Pro-B cell lines.** Wild type RAG2 Thy1.1 retroviral plasmid was a gift from Barry Sleckman lab[67]. $Rag2^{-/-}Xrcc4^{-/-}$ $Polq^{-/-}$ $p53^{-/-}$ v-Abl pro-B cells were complemented by retroviral infection and Thy1.1$^+$ cells were purified using magnetic separation (Miltenyi 130-121-273). Equal number of Thy1.1$^+$ ($Rag2^+$) and Thy1.1$^-$ ($Rag2^{-/-}$) cells were mixed, treated with ABLki and

released from ABLki. The percentage of live Thy1.1$^+$ and Thy1.1$^-$ cells were then quantified by FACS at different time point. Samples without ABLki treatment were used as controls. Human XRCC4 cDNA was cloned into plasmid MSCV-IRES-Thy1.1 DEST (addgene #17442) to produce Xrcc4-Thy1.1 retrovirus. XRCC4-deficient v-Abl pro-B cells were infected by XRCC4-Thy1.1 retrovirus and Thy1.1$^+$ cells were purified using magnetic separation (Miltenyi 130-121-273).

**Cas9-induced DSB repair assay.** For *Bcr-Abl* translocation induction, $5.10^6$ cells were nucleofected (Amaxa 4D, Solution P1, Program DO-100) with the two sgRNAs and the pCas9-GFP protein (ratio 2:1) (see Supplementary Table 2 for sgRNAs sequences). For cycling cell conditions, unsynchronized pro-B cells were harvested 48 h after transfection and genomic DNA was extracted. Non-transfected cells were used as controls. For G1-arrested and released cell conditions, pro-B cells were incubated with the ABLki for 40 h (i.e., time to arrest cells in G1), transfected with the sgRNA/Cas9 protein mixture and kept in ABLki for an additional 48 h. Half of the G1-arrested cells were harvested for genomic DNA extraction (G1 blocked condition). The other half of the cells were washed twice with PBS and cultured in normal media for 5 additional days before genomic DNA extraction (released condition). Translocation was detected by PCR on 50 ng DNA. For sequencing of translocation breakpoints, PCR products were cloned in TOPO TA vector following manufacturer instructions. Translocation frequency was assessed using direct PCR on DNA dilutions from 50 ng to 0,37 ng as previously described[41]. *Bcr* and *Abl* endogenous breakpoint sites were sequenced and analyzed using « TIDE: Tracking of Indels by DEcomposition » webtool (https://tide.deskgen.com/).

**Western blotting.** Cells were lysed using RIPA cell lysis reagent (Thermo Fisher Scientific 89900) and protease inhibitors cocktail (Roche 11873580001). Equal amounts of proteins were subjected to SDS-PAGE on 4–12% Bis-Tris gel. Proteins were transferred onto a nitrocellulose membrane (Life Technologies) using the iBlot apparatus (P3 program, 7 min transfer, Invitrogen). For p53 and PARP1 blots, the membranes were incubated in 5% non-fat dried milk in TBS containing 0.1% Tween-20 buffer for at least 1 h at room temperature, and subsequently incubated overnight at 4 °C with primary antibody (p53: Santa Cruz sc-393031, clone A-1, 1:2000; PARP1: clone A6.4.12; 1:50). γ-Tubulin (Sigma Aldrich T6557, clone GTU-88, 1:10000) or β-Actin (Sigma Aldrich A1978, clone AC-15, 1:5000) was used as a loading control. HRP-conjugated anti-mouse secondary antibody (Cell Signaling Technology 7076, 1:10000) was used. Immune complexes were detected with Clarity Western ECL Substrate (Biorad) or WesternBright Sirius substrate (Advansta). Blots were developed and analyzed using Image Lab v6.0 (Biorad). For XRCC4 blots, the membranes were incubated in Intercept (TBS) Blocking Buffer (LI-COR) for 1 h at room temperature, and incubated successively with XRCC4 (Santa Cruz sc-8285, clone C20, 1:1300) and γ-Tubulin primary antibodies. IRDye-conjugated anti-goat (LI-COR IRDye® 800CW Donkey anti-Goat IgG, 926-32214, 1:10,000) and anti-mouse (LI-COR IRDye® 680RD Goat anti-Mouse IgG, 926-68070, 1:20,000) secondary antibodies were used. Immune complexes were detected using LI-COR Odyssey Imaging System. For Cas9-GFP western blot, G1 blocked cells were nucleofected with pCas9-GFP protein and cultured with ABLki for 72 additional hours. Proteins were extracted at 24, 48 and 72 h. Whole cell extracts were prepared with protein lysis buffer (50 mM Tris-HCl at pH 7.4, 1%Triton X-100, 0.1% SDS, 150 mM NaCl, 1 mM EDTA, and 1 mM DTT), with addition of Complete cocktail protease inhibitor tablets (Roche). Typically, 20 μg of protein extracts were run on an 8% (w/v) Tris-HCl SDS PAGE gel. Gel was blotted and probed with GFP antibody (Santa Cruz sc-9996, clone B-2, 1:1000). Vinculin antibody (Santa Cruz sc-73614, clone 7F9, 1:1000) was used as a loading control. Antibodies were used according to manufacturer's recommendations. Secondary antibodies were IRdye-conjugated (LI-COR IRDye® 800CW Goat anti- Mouse IgG, 926-32210, 1:10000 for GFP and IRDye® 680RD Goat anti-Mouse IgG, 926-68070, 1:10000 for Vinculin), and blot developed with LI-COR Odyssey CLX imaging system.

**LAM-HTGTS.** Linear amplification–mediated high-throughput genome-wide translocation sequencing (LAM-HTGTS) was performed according to a published protocol[40]. Briefly, DNA was purified and sonicated (Bioruptor, Diagenode) into 500–1,000 bp fragments. LAM-PCR was performed using bait (IgkJ4) primer coupled with biotin, 1U Phusion polymerase (ThermoFisher Scientific F530L) and the following cycles: 1× (98 °C for 120 s; 80× (95 °C for 30 s; 58 °C for 30 s; 72 °C for 90 s); and 1× (72 °C for 120 s). Biotinylated PCR fragments were incubated with MyOne streptavidin C1 beads (Invitrogen 65001) and rotated for 4 h in 1 M NaCl and 5 mM EDTA buffer at room temperature. After washes with B&W buffer (1 M NaCl, 5 mM Tris-HCl (pH 7.4) and 0.5 mM EDTA (pH 8.0)), on-bead ligation was performed using 2.5 μM bridge adapter, 1 mM hexamine cobalt chloride (Sigma H7891), 15 U T4 DNA ligase (Promega M1804), 15% PEG-8000 (Sigma P2139) and the following cycles: 25 °C for 1 h; 22 °C for 2 h; and 16 °C O/N. After washing three times with B&W buffer, the on-bead ligated products were subjected to nested PCR using Phusion polymerase, locus-specific (IgkJ4 or *c-myc*) and adapter primers and the following cycles: 1× (95 °C for 300 s); 15x (95 °C for 60 s; 60 °C for 30 s; 72 °C 60 s); and 1× (72 °C 600 s). Blocking digestion was then performed with 5U *EcoNI* (NEB R0521S; for IgkJ4) *or* *PvuII* (NEB R3151S; for *c-myc*) for 1 h to remove uncut germline DNA. After purification using the QIAquick Gel Extraction Kit (Qiagen 28704), recovered DNA was PCR-amplified using Illumina primers, Phusion polymerase and the following cycles: 1× (95 °C for 180 s); 10× (95 °C for 30 s; 60 °C for 30 s; 72 °C for 60 s); and 1× (72 °C for 360 s). The tagged PCR products were size-selected for DNA fragments between 500–1000 bp on a 1% agarose gel and purified using a QIAquick Gel Extraction Kit before loading onto an Illumina Miseq machine for paired-end 2 × 250 bp sequencing.

***Igk* locus capture-based target enrichment for sequencing.** Coding and signal sequences of 140 mouse *Igk* V and 5 Igk J segments were downloaded from MGI database. 280 bps windows flanking recombination signal site (RSSs) of Igk V and J segments, including coding and signal sequences, were used to design Agilent SureSelect customized probes (Agilent Technologies). A total of 2348 120-mer probes (1168 for signal ends; 1180 for coding ends) were designed for *Igk* locus capture. SureSelect$^{XT}$ Target Enrichment System kit (Agilent Technologies) was used to prepare *Igk* locus capture sequencing libraries. Briefly, 3 μg gDNA was sonicated to 150 to 200 bps size. Sheared DNA was end repaired, dA tailed, ligated with provided paired-end adapter and amplified. The prepared DNA was hybridized to customized probes and captured by Dynabeads MyOne Streptavidin T1 (Life Technologies). The pull-downed DNA was amplified with provided indexing primers of Illumina MiSeq for 250 bps paired-end sequencing.

**Data analysis and visualization.** LAM-HTGTS (see also Supplementary Fig. 5). Sequencing reads in FASTQ files were decoded and adapter sequence trimmed using by ea-utils tool (https://expressionanalysis.github.io/ea-utils/). Paired-end reads with less than 20 nucleotides bait sequence in the 5' end of read1 were filtered out. Paired-end reads were processed by Samtools and mapped to the mouse mm9 reference genome using BWA mem[68,69] with default settings. Paired-end reads with a mapping quality less than 10 and multiple mapped reads were filtered out. Mapped reads were then further analyzed using read 2. Among mapped read2, those that were not split mapped and mapped within the IgkJ4 bait to IgkJ2 regions were defined as germline. Among non-split mapped and non-germline read2, those with more than three quarters length of mapped sequence were defined as mapped with >75% sequence reads. Split mapped read2 that partly and uniquely mapped within repetitive genomic regions were categorized into uniquely mapped repeat reads. Split mapped read2 with part of sequence extending beyond IgkJ4 recombination signal site were defined as split joins with IgkJ1 or J2 since the breakpoint likely originated from IgkJ1 or J2 RSS cutting followed by extensive resection. Split mapped read2 with part of sequence mapped between IgkJ4 RSS and bait primer coordinate were defined as split mapping joins with IgkJ4 bait. Split mapping joins with IgkJ4 bait were further classified according to prey genomic location: trans-joins with antigen receptor (AgR) genes (*Igh, Igl, Tcrα, Tcrδ, Tcrγ, Tcrβ*), cis-joins with Igk V region and trans-joins with non AgR loci.

Only the reads with split joins were further considered for microhomology and resection length analysis. For the calculation of resection length, the annotated *Igk* V and J segments were downloaded from MGI database. Only the segments with annotated recombination signal site were considered. To precisely calculate the resection length, V segments that are close to each other (less than 4 kbps distance) were filtered out and only 144 *Igk* V segments were kept. The resection toward both coding and signal end were calculated and the maximum resection length was set up to 4 kbps.

For data analysis of *Igk* pull-down libraries, 2 × 250 bps paired-end reads were firstly assembled by FLASH tool[70] by at least 10 nts overlapping. Both assembled and un-assembled reads were mapped to mouse mm9 reference genome using BWA mem[68,69] using default settings. Reads with mapping quality less than 10 and multiple mapping were filtered out. Reads containing *Igk* sequence were disentangled by Bedtools[71] through intersection with *Igk* locus coordinate. These reads were further dissected into the following categories: *Igk* V-V joins, V-J joins and germlines by costumed perl scripts. To avoid cryptic *Igk* V-V joins, the cutoff distance between 2 split reads for *Igk* V-V joins was set to >2 kbps. The joins with distances <2 kbps were defined as short deletion. To visualize the *Igk* V-J joins, *Igk* locus was divided into bins with 100 bps size.

Circos[72] was used to visualize chromosomal rearrangements within each 1kbp bin. IgkJ4-$V_{region}$ junctions in *Igk* loci were visualized by IGV v2.4.5.

The statistical analysis was performed by R v3.5.1 and visualized by RStudio v1.1.463, Graph Prism v7.04 and Microsoft Excel v16.16.2.

**Reporting summary.** Further information on research design is available in the Nature Research Reporting Summary linked to this article.

## Data availability

The LAM-HTGTS and *Igk* capture sequencing data have been deposited in NCBI's Gene Expression Omnibus under accession number GSE138137. Coding and signal sequences of mouse Igk V and Igk J segments were downloaded from MGI database (http://www.informatics.jax.org/). Repetitive genomic coordinate for mouse reference genome were downloaded from NCBI Table Brower (http://genome.ucsc.edu/cgi-bin/hgTables). The source data for figures and tables are provided in Source Data file. Source data are provided with this paper.

## Code availability

Custom scripts have been deposited in https://github.com/yuwei4891/DerianoLab-HTGTS.git.

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

## Acknowledgements

We thank the Institut Pasteur genomics and cytometry platforms for help with sequencing and cell sorting, Frederick Alt and members of his lab for tremendous help with LAM-HTGTS experiments, Barry Sleckman for providing the RAG2-Thy1.1 complementation vector, Joy Bianchi for providing the *Rag2⁻/⁻ p53⁻/⁻ v-Abl* pro-B cell lines, Carine Giovannangeli and Anne de Cian for providing the Cas9 protein and Thomas Mercher for helpful discussions and suggestions. This project is funded by the Institut Pasteur (L.D. and C.D. labs), the Institut National du Cancer (INCa Grant # PLBIO16-181 to L.D. and E.B. labs), the Ligue Nationale Contre le Cancer (Équipe Labellisée 2019 L.D. lab and Équipe Labellisée 2017 and 2020 E.B. lab) the Cancéropôle IdF-INCa (Emergence 2016 grant to L.D. and C.D.), the Spanish Ministerio de Economía, Industria y Competitividad (Grant SAF2017-83565-R to J.Y.) as well as by the Fundación Científica de la Asociación Española Contra el Cáncer (AECC) (Grant PROYEI6018YÉLA to J.Y.).

## Author contributions

L.D. conceived the study and supervised the work. L.D., W.Y., C.L., L.B. and E.B. designed experiments, analyzed data, discussed interpretation of the results and wrote the manuscript. W.Y. performed experiments with contribution from C.L., L.B., M.B.-F. and H.L.-H. C.L. adapted LAM-HTGTS protocol. L.B. performed Cas9-induced *Bcr-Abl* translocation experiments. M.B.-F. and H.L.-H. helped with generation and characterization of knock out cell lines. Lu.Ba. and C.D. helped established the leukemia mouse model protocol. W.Y. performed in vivo mouse leukemia experiments and analysis with help from C.L. and M.B.-F. W.Y. performed LAM-HTGTS sequencing and analysis with help from C.L. C.L., L.B. and J.Y. performed Western Blots. W.Y., C.L., M.B.-F. and H.L.-H. performed cytogenetic analysis. J.Y. performed PARP activity assays. All authors read and approved the manuscript.

## Competing interests

The authors declare no competing interests.
