## [Peer Review File · Nature Communications]

REVIEWER COMMENTS

Reviewer #1 (Remarks to the Author):

While the repair of DNA double strand breaks (DSB)s occurs predominantly either by homologous recombination or non-homologous end joining, there are other less-well defined minor pathways, often referred to as as alternative NHEJ. Interest in alt NHEJ has been enhanced by the evidence that it is a promising therapeutic target in homologous recombination-deficient tumors. In this study, the authors examine the contribution of alt NHEJ to the repair of DNA double strand breaks that occur in the G1 phase of the cell cycle in NHEJ-deficient cells. Specifically, they show that DSBs induced during G1 are repaired in an error-prone manner by alt NHEJ during the S and G2 phases. This provides novel mechanistic insights as to how alt NHEJ serves as a back-up for NHEJ and evidence that alt NHEJ is a promising target for NHEJ-deficient cancers. The experimental approach is elegant and the results support the author's conclusions.

There are a few minor concerns;

The last sentence in the Summary should be revised- while the study does show that G1 DSBs are a source of genome instability if not repaired by NHEJ, I don't think that it is correct to say that they are a novel therapeutic target.

In the first part of the discussion, the authors should state explicitly that their results are in mouse cells given the apparent differences in the contribution of DSB repair pathways between mouse and human cells.

There are several published studies concluding that PARP1 is a key alt NHEJ factor. The authors should discuss the possible reasons for the discrepancy between these studies and the results of this study showing that PARP1 is dispensable for the repair of G1 DSBs later in the cell cycle by alt NHEJ.

No immunoblotting data showing that the targeted protein is absent in knockout cells

Reviewer #2 (Remarks to the Author):

The manuscript by Yu et al elucidates on the mechanisms by which G1-induced DNA double strand breaks (DSBs) are repaired during the S-G2/M phase of the cell cycle. The authors cleverly employ a progenitor (pro-B) cell line transformed with the v-abl kinase, that allows control over the induction of DSBs by the endonuclease RAG during G1 arrest, and release of cells into S-G2 upon removal of the kinase inhibitor imatinib. This is a timely and well-designed study, that uses state-of-the-art methodology for the analysis of V(D)J recombination products and associated chromosomal translocations (LAM-HTGTS method) in a panel of pro-B cell lines. The findings are sound and support a role for the DNA polymerase theta in the repair of G1-induced DSBs in B-cells deficient for the NHEJ factor XRCC4 and the G1/S checkpoint protein p53. The authors go on to show a synthetic lethal phenotype in cells with combined deficiency in DNA polymerase theta and XRCC4, including using an in vivo model of acute lymphoblastic leukemia. There is interest in understanding the mechanisms by which DNA polymerase theta is involved in alternative NHEJ, given its promising role as a therapeutic target in human cancer. These studies provide an important first step in elucidating the role of this DNA repair factor in V(D)J recombination and leukemia development, and therefore I find them suitable for publication in Nature Communications.

Please find below a few points that require clarification:

1. It would be important to further explore the nature of the few Igk rearrangements present in XRCC4 -/- p53 -/- pro-B cells + ABLki. Igk rearrangements recovered from XRCC4 -/- pro-B cells + ABLki contain a considerable extent of MH (Fig. S7C) and end processing (Fig. 2B), which suggest alternative NHEJ mechanisms can occur in G1 to some extent. How do IgkJ4-to-V junctions look like with regards to end-resection in G1-arrested XRCC4 -/- p53 -/- pro-B cells?

Presumably, alternative NHEJ in XRCC4 -/- p53 -/- pro-B cells + ABLki would not rely on similar mechanisms as observed upon cell cycle release (i.e DNA polymerase theta).

2. Previous studies highlight the presence of templated DNA insertions as a unique feature of DNA polymerase theta-dependent DNA repair mechanisms. Can the authors use this to further discriminate between Igk junctions in XRCC4 -/- pro-B cells + ABLki and XRCC4 -/- p53 -/- pro-B cells +/- ABLki? This may be complicated by the presence of TdT activity, but would further strengthen the point that DNA polymerase theta-dependent alternative NHEJ mechanisms require cell cycle entry.

3. In the context of class switch recombination, deletion of DNA polymerase theta associates with an increased level of Myc/IgH translocations in mature B-cells (Yousefzadeh et al, 2014). These results have led to the conclusion that DNA polymerase theta is part of a sub-pathway of alternative NHEJ that prevent the formation of chromosomal translocations. In Fig.4 the authors find the opposite result in PolQ -/- pro-B cells, which show an increased proportion of chromosomal breaks. Is this phenotype RAG-dependent, or is it also observed when DSBs are generated by Cas9? Addressing this point would help clarify the apparently opposing functions of DNA polymerase theta in chromosomal translocations resulting from different types of immunoglobulin gene rearrangements.

4. Note that recent studies using CRISPR screens to detect synthetic lethal interactions with DNA polymerase theta, failed to identify XCCR4 as a significant target (Feng et al, 2019). The authors should address this point in the discussion and provide an explanation for this apparent discrepancy.

Reviewer #3 (Remarks to the Author):

In this study the authors used a very elegant system using transformed pro-B cells to generate DSBs (either through Rag1/2 or with Cas9) in G1-arrested cells in order to investigate the fate of DSBs induced in G1 throughout the cell cycle. The authors show that DSBs induced in G1 by Rag1/2 or Cas9 cannot be repaired in an NHEJ-deficient background (Xrcc4-/-). By knocking out p53, cells are then able to bypass the G1/S checkpoint and the authors show that upon cell cycle re-entry, G1-induced DSBs are repaired by an error prone alternative end-joining pathway that is associated with extensive DNA end resection, microhomology and which leads to substantial genomic instability and chromosomal translocations. Surprisingly, the authors show that the repair of G1-induced DSBs in Xrcc4-/- p53-/- cells is independent of Parp1, but rather relies on the induction of Polq, which suppresses the accumulation of unresolved DNA ends during cell cycle progression. The authors then show that Xrcc4 and Polq display synthetic lethality to Rag1/2 induced DSBs both in vitro and in vivo, thus opening the possibility to use Polq as a therapeutic target to eradicate NHEJ-deficient cancer cells in combination with DSB-inducing agents.

The manuscript addresses a very relevant topic, is timely and I am very enthusiastic about the work and conclusions.

I have only a few minor comments:

A quick explanation of the different cell lines used should be included in the figure legends (i.e. Xrcc4-/- p53-/- (Xr15307-3 & Xr15307-11)).

The gRNAs used to knockout XRCC4, p53, Polq, Rag2, should be added to supplementary table S2.

A western blot for Xrcc4, p53, Parp1, Polq for the different knockout cell lines would have been a good addition.

Why are there IgkV10-95-J4 rearrangements in the non-treated p53-/- cells in Figure 1C ?

The Parp activity measurements are not very convincing and the figure doesn't really reflect what's written in the results section. For me, "the PARP activity was not dramatically reduced in PARP1-deficient B cell clones"... This should be tuned down.

Concerning the Cas9-induced DSBs in G1 arrested cells, I wonder what is the half-life of the Cas9-GFP protein ? Was the absence of Cas9-GFP verified by flow cytometry or western blot before release ? If not, it is possible that Cas9 also induced DSBs in S/G2.

August 03, 2020

Response to the reviewers' comments

We are grateful to the referees for their careful review of the manuscript, for their positive comments about our work and for insightful comments and suggestions for improving our paper. After a disturbing time, including closing of all labs involved in this study and additional measures related to the SARS-CoV-2 pandemic that limited our ability to resume work in the laboratories, we are glad to be able to submit our revised manuscript by Yu et al. that includes additional controls and new experiments as well as clarifications and further discussion of some of our findings. What follows is a point-by-point response (in blue) to the reviewer's comments (in black).

Reviewer #1:

While the repair of DNA double strand breaks (DSB)s occurs predominantly either by homologous recombination or non-homologous end joining, there are other less-well defined minor pathways, often referred to as alternative NHEJ. Interest in alt NHEJ has been enhanced by the evidence that it is a promising therapeutic target in homologous recombination-deficient tumors. In this study, the authors examine the contribution of alt NHEJ to the repair of DNA double strand breaks that occur in the G1 phase of the cell cycle in NHEJ-deficient cells. Specifically, they show that DSBs induced during G1 are repaired in an error-prone manner by alt NHEJ during the S and G2 phases. This provides novel mechanistic insights as to how alt NHEJ serves as a back-up for NHEJ and evidence that alt NHEJ is a promising target for NHEJ-deficient cancers. The experimental approach is elegant and the results support the author's conclusions.

There are a few minor concerns;

The last sentence in the Summary should be revised- while the study does show that G1 DSBs are a source of genome instability if not repaired by NHEJ, I don't think that it is correct to say that they are a novel therapeutic target.

We have modified the last sentence of the summary: "Our findings provide a mechanistic framework for the repair of G1 DSBs progressing to S-G2/M and support the conclusion that this type of DNA damage repair event drives genomic instability in cancer cells and represent an attractive target for future DNA repair-based therapies".

In the first part of the discussion, the authors should state explicitly that their results are in mouse cells given the apparent differences in the contribution of DSB repair pathways between mouse and human cells.

We agree with Reviewer #1 and have clarified this point in the discussion: "Using a murine experimental system in which we can induce reversible cell cycle arrest and apply robust DSB joining assays, we find that in the absence of XRCC4, end joining is virtually absent in G1-phase cells...". We also discuss the possible implications of our findings in human cells in the discussion section: " In contrast, we have previously shown that breakpoint junctions ...".

There are several published studies concluding that PARP1 is a key alt NHEJ factor. The authors should discuss the possible reasons for the discrepancy between these studies and the results of this study showing that PARP1 is dispensable for the repair of G1 DSBs later in the cell cycle by alt NHEJ.

We thank reviewer #1 for pointing this out. We have now included a specific discussion of our results in the context of the published literature on PARP1, Pol θ and alt-NHEJ (see also response to reviewer #2's comment #3).

No immunoblotting data showing that the targeted protein is absent in knockout cells

We now provide a Western blot for XRCC4, p53 and PARP1 for the corresponding knockout pro-B cell lines (Figures S1B, C, D). Unfortunately, to our knowledge, there is no specific antibody to detect murine Pol θ currently available. Instead, we now provide PCR-based genotyping data (gels and sequence information) for all *Polq*^{-/-} pro-B cell lines used in this study (Figure S1A and Table S1).

Reviewer #2:

The manuscript by Yu et al elucidates on the mechanisms by which G1-induced DNA double strand breaks (DSBs) are repaired during the S-G2/M phase of the cell cycle. The authors cleverly employ a progenitor (pro-B) cell line transformed with the v-abl kinase, that allows control over the induction of DSBs by the endonuclease RAG during G1 arrest, and release of cells into S-G2 upon removal of the kinase inhibitor imatinib. This is a timely and well-designed study, that uses state-of-the-art methodology for the analysis of V(D)J recombination products and associated chromosomal translocations (LAM-HTGTS method) in a panel of pro-B cell lines. The findings are sound and support a role for the DNA polymerase theta in the repair of G1-induced DSBs in B-cells deficient for the NHEJ factor XRCC4 and the G1/S checkpoint protein p53. The authors go on to show a synthetic lethal phenotype in cells with combined deficiency in DNA polymerase theta and XRCC4, including using an in vivo model of acute lymphoblastic leukemia. There is interest in understanding the mechanisms by which DNA polymerase theta is involved in alternative NHEJ, given its promising role as a therapeutic target in human cancer. These studies provide an important first step in elucidating the role of this DNA repair factor in V(D)J recombination and leukemia development, and therefore I find them suitable for publication in Nature Communications.

Please find below a few points that require clarification:

1. It would be important to further explore the nature of the few Igk rearrangements present in XRCC4^{-/-} p53^{-/-} pro-B cells + ABLki. Igk rearrangements recovered from XRCC4^{-/-} pro-B cells + ABLki contain a considerable extent of MH (Fig. S7C) and end processing (Fig. 2B), which suggest alternative NHEJ mechanisms can occur in G1 to some extent. How do IgkJ4-to-V junctions look like with regards to end-resection in G1-arrested XRCC4^{-/-} p53^{-/-} pro-B cells? Presumably, alternative NHEJ in XRCC4^{-/-} p53^{-/-} pro-B cells + ABLki would not rely on similar mechanisms as observed upon cell cycle release (i.e DNA polymerase theta).

We thank Reviewer #2 for pointing this out.

As shown in the original manuscript, there is a very low level of joining (*i.e.* Igk rearrangements) in ABLki-treated G1-arrested *Xrcc4*^{-/-} pro-B cells (431 *Jk₄-V_{k_s}* joins corresponding to 0.016 % of total Igk mapped reads) (Table S4A). These joins differ from Igk joins observed in wild type cells as they typically lack nucleotide insertions (7.5% in *Xrcc4*^{-/-} cells versus 32.2% in wild type cells) (Figure S8B), contain more frequently MHs (2-to-6bp MH: 43.3% in *Xrcc4*^{-/-} cells versus 17.3% in wild type

cells) (Figure S8C) and suffer increased DNA end processing (mean resection = 7.5 bp in *Xrcc4*^{-/-} cells versus 3.5 bp in wild type cells) (Figure 2B). *Thus, we agree with reviewer #2 that alternative NHEJ can occur in G1 XRCC4-deficient pro-B cells, although to some extremely limited extent.*

We also noticed that the length of resection in G1-arrested XRCC4-deficient pro-B cells is much smaller than the one we observe in released *Xrcc4*^{-/-} *p53*^{-/-} pro-B cells (mean resection length = 730.1 bp) (Figure 2C), possibly reflecting the action of anti-resection mechanisms in G1 cells (e.g. 53BP1 and its downstream effectors, the Ku heterodimer, the RAG post cleavage complex and possibly others). As Pol θ expression is limited to the S, G2/M phases of the cell cycle, *alternative NHEJ in G1 is distinct from alternative NHEJ operating upon cell cycle release with regards to the efficacy of the repair pathway (i.e. low versus high), the type of repair products (i.e. short versus long resection) and the factors involved (i.e. pol θ independent versus pol θ dependent).*

We agree with reviewer #2 that the analysis of end resection in G1-arrested *Xrcc4*^{-/-} *p53*^{-/-} pro-B cells would strengthen these conclusions. We did perform such analysis and found that rare *Igk*J_κ-to-*Vk_s* joins recovered from G1-arrested *Xrcc4*^{-/-} *p53*^{-/-} pro-B cells (484 *Jk_κ*-*Vk_s* joins corresponding to 0.029 % of total *Igk* mapped reads) (Table S4A) harbor long DNA end resection (mean resection = 347.3 nucleotides) (see **Author response Figure 1** below), albeit to a lesser extent than released cells (mean resection length in released *Xrcc4*^{-/-} *p53*^{-/-} pro-B cells = 730.1 nt with 162,649 *Jk_κ*-*Vk_s* joins corresponding to 3.41 % of total *Igk* mapped reads) (Figure 2C and Table S4A). This was rather surprising as the few rearrangements recovered from G1-arrested *Xrcc4*^{-/-} pro-B cells contained minimal resection (7.5 nt). We reasoned that, in the absence of p53 checkpoint, sporadic RAG expression in untreated cycling *Xrcc4*^{-/-} *p53*^{-/-} pro-B cell clones might lead to low levels of *Igk* rearrangements in these cells prior to ABLki treatment. In the context of proliferating XRCC4/p53 doubly deficient cells, we reasoned that a majority of these sporadic RAG-induced DSBs in G1 cells would progress to S-G2-M and be repaired by S-phase alternative NHEJ, leading to extensive end resection. This background level of *Igk* rearrangement in untreated cells would then confound the analysis of de novo G1 *Igk* DSB repair products generated in ABLki-treated G1-arrested *Xrcc4*^{-/-} *p53*^{-/-} cells. To test such scenario, we generated and sequenced two novel *Igk* *Jk_κ* HTGTS libraries from untreated *Xrcc4*^{-/-} *p53*^{-/-} cells. As anticipated, sequence analysis revealed low levels of *Igk* *Jk_κ*-*Vk_s* rearrangements (387 *Jk_κ*-*Vk_s* joins out of 2,621,24 total *Igk* mapped reads; 0.015%) in these cells with joints harboring extensive resection (mean resection length = 482.8 nucleotides) (see **Author response Figure 1** below). Thus, although theoretically informative, a thorough analysis of de novo G1-induced *Igk* rearrangements in *Xrcc4*^{-/-} *p53*^{-/-} pro-B cells is compromised by the presence of rearrangements generated in proliferative conditions prior to ABLki treatment. *Because of these imperfect experimental conditions and the very low number of events that are recovered in these cells, we have decided to show these results to the response to the reviewers only and leave them out of the manuscript.*

Author response Figure 1. (*N.B.*: To measure end resection events most accurately, we only analyzed annotated *IgkV* recombining segments that localize more than 4kb one from another, hence the number of reads analyzed for calculating the resection length is lower than the total number of reads mapping within the *Igk* locus).

2. Previous studies highlight the presence of templated DNA insertions as a unique feature of DNA polymerase theta-dependent DNA repair mechanisms. Can the authors use this to further discriminate between *Igk* junctions in *XRCC4*^{-/-} pro-B cells + ABLki and *XRCC4*^{-/-} *p53*^{-/-} pro-B cells +/- ABLki? This may be complicated by the presence of TdT activity, but would further strengthen the point that DNA polymerase theta-dependent alternative NHEJ mechanisms require cell cycle entry.

We provide the % of *Igk* junctions with nucleotide insertions in Figure S8B: 7.5% in G1-blocked *Xrcc4*^{-/-} pro-B cells; 13.1% in G1-blocked *Xrcc4*^{-/-} *p53*^{-/-} pro-B cells and 18.2% in released *Xrcc4*^{-/-} *p53*^{-/-} pro-B cells. For the abovementioned reasons, *Igk* joints recovered from G1-blocked *Xrcc4*^{-/-} *p53*^{-/-} pro-B cells correspond to a mix of junctions originating from cell culture and G1-arrested conditions and thus cannot be used to compare DNA insertion profiles in G1-blocked and released conditions. Nevertheless, when comparing *Igk* junctions recovered from G1 *Xrcc4*^{-/-} B cells (*i.e.* pol θ-independent) and *Igk* junctions recovered from released *Xrcc4*^{-/-} *p53*^{-/-} B cells (*i.e.* pol θ-dependent), we do see an increase in the % of insertions at *Igk* junctions that might represent templated DNA insertions (7.5% and 18.2%, respectively).

In addition, we do agree with reviewer #2 that this type of analysis is complicated by the presence of TdT activity in pro-B cell lines. This is an observation that we made in a previous study when comparing the presence of nucleotide insertions in human B cell lines and human fibroblasts (Ghezraoui H. et al. Mol Cell, 2014, <https://doi.org/10.1016/j.molcel.2014.08.002>).

3. In the context of class switch recombination, deletion of DNA polymerase theta associates with an increased level of Myc/IgH translocations in mature B-cells (Yousefzadeh et al, 2014). These results have led to the conclusion that DNA polymerase theta is part of a sub-pathway of alternative NHEJ that prevent the formation of chromosomal translocations. In Fig.4 the authors find the opposite result in *PolQ*^{-/-} pro-B cells, which show an increased proportion of chromosomal breaks. Is this phenotype RAG-dependent, or is it also observed when DSBs are generated by Cas9? Addressing this point would help clarify the apparently opposing functions of DNA polymerase theta in chromosomal translocations resulting from different types of immunoglobulin gene rearrangements.

There is a confounding literature on the specific role of Pol θ in the generation of chromosomal translocations. As mentioned by reviewer #2, in the context of class switch recombination in NHEJ-proficient mature B cells, deletion of DNA polymerase theta associates with an increased level of Myc/IgH translocations (Yousefzadeh et al., PLoS Genet, 2014), indicating that Pol θ suppresses genetic instability. In support to this, Wyatt et al. found that deletion of Pol θ promotes chromosomal translocations at Cas9-DSBs in transformed mouse embryonic fibroblasts, although a significant effect was limited to cells already deficient in Ku70 (Wyatt et al., Mol Cell, 2016). In contrast, Mateo-Gomez et al. found that deletion of Pol θ associates with a decrease level of Cas9-induced chromosomal translocations in mouse pluripotent cells and, similarly, a decrease level of telomere fusions in mouse embryonic fibroblasts in which the shelterin complex and Ku80 have been depleted (Mateo-Gomez et al., Nature, 2015). These studies indicate that Pol θ both suppresses and promotes

DSB-induced chromosomal translocations and that this discrepancy may reflect differences in the origin of the chromosome break, cell type and possibly other biological parameters.

To address this important issue, we now provide a complete analysis of Cas9-induced chromosomal translocations in Pol θ -proficient and Pol θ -deficient cells using both PCR-based and cytogenetic-based chromosomal translocation measurements. We believe our new results help clarify the current literature with regard to the role of Pol θ in the generation of genetic instability and highlight the importance of the cell cycle in Pol θ -dependent versus Pol θ -independent alt NHEJ activities. From these experiments, we conclude that:

1) As with the RAG enzyme, Cas9-induced DSBs very rarely generate chromosomal translocations in G1-arrested cells.

2) As with the RAG enzyme, Cas9-induced DSBs in G1-arrested XRCC4/p53-deficient pro-B cells promote *Bcr-Abl* chromosomal translocations upon release into S-G2/M in a manner that is dependent on Pol θ . These results strengthen one of the main conclusions of our paper that is, *Pol θ is essential for alt-NHEJ of unrepaired G1 DSBs progressing to S-G2/M, promoting chromosomal translocations and preventing the accumulation of broken chromosomes during cell cycle transition to M phase.*

3) In cycling *Xrcc4^{-/-} p53^{-/-}* pro-B cells, Cas9-induced DSBs promote chromosomal translocations through both Pol θ -dependent and Pol θ -independent alt NHEJ. In normal cell culture conditions, approximately 60% of the cells are in the S-phase of the cell cycle, the remaining being mostly in G1 (Figure 1B and Figure S3). Thus, in the absence of XRCC4 and p53, *as opposed to DSBs induced in G1, DSBs generated in S can lead to chromosomal translocations in a manner that is independent of Pol θ .*

4) In agreement with work from Yousefzadeh et al., we measured by cytogenetics a two- to three-fold increase in Cas9-induced chromosomal translocations in cycling *Polq^{-/-} p53^{-/-}* pro-B cells as compared to cycling *p53^{-/-}* (3.2% and 1.35%, respectively), although this difference could not be reproducibly detected in our PCR-based translocation assay. In addition, in cycling cells, chromosomal translocations were always increased in XRCC4-deficient cells as compared to XRCC4-proficient cells, regardless of Pol θ deletion. Thus, *Pol θ principally plays a role in the generation of chromosomal translocations in the context of NHEJ deficiency, Pol θ being crucial for the error-prone joining of unrepaired G1 DSBs progressing to S-G2/M.*

These data are presented in Figure S13 and discussed in the discussion section of our manuscript.

4. Note that recent studies using CRISPR screens to detect synthetic lethal interactions with DNA polymerase theta, failed to identify XCCR4 as a significant target (Feng et al, 2019). The authors should address this point in the discussion and provide an explanation for this apparent discrepancy.

The CRISPR loss of function screens performed in Feng et al detect gene mutations that are lethal in *Polq^{-/-}* cells in normal cell culture conditions (*i.e.* in the absence of genotoxic treatment or induced DSBs). As shown in Figure S12, deficiencies in XRCC4 and Pol θ have synergistic effects on cell growth and viability but do not lead to synthetic lethality. Synthetic lethality requires the induction of G1 DSBs due to the accumulation of unrepaired DSBs during subsequent cell divisions (shown in

Figure 5). Thus, it is possible that the CRISPR screens performed by Feng et al. lack sensitivity to identify XRCC4 as a significant target.

Consistent with the possible lack of sensitivity of the screens, the authors also failed to identify BRCA1 as a significant target, although recent publications have reported a synthetic lethal interaction between BRCA1 and Pol θ (Mateo-Gomez et al., Nature, 2015; Ceccaldi et al., Nature 2015).

Reviewer #3:

In this study the authors used a very elegant system using transformed pro-B cells to generate DSBs (either through Rag1/2 or with Cas9) in G1-arrested cells in order to investigate the fate of DSBs induced in G1 throughout the cell cycle. The authors show that DSBs induced in G1 by Rag1/2 or Cas9 cannot be repaired in an NHEJ-deficient background (*Xrcc4*^{-/-}). By knocking out p53, cells are then able to bypass the G1/S checkpoint and the authors show that upon cell cycle re-entry, G1-induced DSBs are repaired by an error prone alternative end-joining pathway that is associated with extensive DNA end resection, microhomology and which leads to substantial genomic instability and chromosomal translocations. Surprisingly, the authors show that the repair of G1-induced DSBs in *Xrcc4*^{-/-} *p53*^{-/-} cells is independent of Parp1, but rather relies on the induction of Polq, which suppresses the accumulation of unresolved DNA ends during cell cycle progression. The authors then show that *Xrcc4* and Polq display synthetic lethality to Rag1/2 induced DSBs both in vitro and in vivo, thus opening the possibility to use Polq as a therapeutic target to eradicate NHEJ-deficient cancer cells in combination with DSB-inducing agents.

The manuscript addresses a very relevant topic, is timely and I am very enthusiastic about the work and conclusions.

I have only a few minor comments:

A quick explanation of the different cell lines used should be included in the figure legends (i.e. *Xrcc4*^{-/-} *p53*^{-/-} (Xr15307-3 & Xr15307-11)).

We thank reviewer #3 for pointing this out and have included this information in the figure legends. In addition, we provide all original data in the source file.

The gRNAs used to knockout XRCC4, p53, Polq, Rag2, should be added to supplementary table S2.

The gRNAs used to knockout *Xrcc4* and *Polq* are listed in Table S2. As indicated in Table S1, *p53*^{-/-} and *Rag2*^{-/-} *p53*^{-/-} pro-B cell lines have been generated from previously published mouse models, *p53*^{-/-} and *Rag2*^{-/-} *p53*^{-/-} mice respectively, and not by CRISPR/Cas9 editing.

A western blot for *Xrcc4*, p53, Parp1, Polq for the different knockout cell lines would have been a good addition.

We now provide a Western blot for XRCC4, p53 and PARP1 for the corresponding knockout pro-B cell lines (Figures S1B, C, D). Unfortunately, to our knowledge, there is no specific antibody to detect murine Pol θ currently available. Instead, we now provide PCR-based genotyping data (gels and sequence information) for all *Polq*^{-/-} pro-B cell lines used in this study (Figure S1A and Table S1).

Why are there IgkV10-95-J4 rearrangements in the non-treated p53^{-/-} cells in Figure 1C ?

As mentioned in our response to reviewer #2 first comment, in the absence of p53, *v-Abl* immortalized pro-B cells have a higher propensity to accumulate low levels of *Igk* rearrangements during normal cell culture conditions. However, in p53-deficient clones, RAG-induced DSBs at the *Igk* locus are repaired by canonical NHEJ in G1 and thus do not interfere with our analysis of de novo RAG-induced *Igk* rearrangements generated upon ABLki treatment, which is robust and represent the vast majority of the joints analyzed in G1-arrested cells. In support to this, analysis of *Igk* joints in G1-arrested and released p53^{-/-} pro-B cells revealed similar levels of end resection (3.2 bp and 3.9 bp, respectively), microhomology (21.6 % and 22.3 %, respectively) and nucleotide insertions (18.2 % and 18.9 %, respectively) (Figure 2 and Figure S8), indicating that RAG-induced *Igk* DSBs were repaired by canonical NHEJ in G1.

The Parp activity measurements are not very convincing and the figure doesn't really reflect what's written in the results section. For me, "the PARP activity was not dramatically reduced in PARP1-deficient B cell clones" ... This should be tuned down.

We agree with reviewer #3 and have modified the sentence accordingly "*As expected*^{d3}, PARP activity was ~~dramatically~~ reduced in PARP1-deficient B cell clones as compared to PARP1-proficient B cell clones". Of note, it is likely that residual PARP activities originate from other PARP enzymes in these cell clones (e.g. PARP2).

Concerning the Cas9-induced DSBs in G1 arrested cells, I wonder what is the half-life of the Cas9-GFP protein? Was the absence of Cas9-GFP verified by flow cytometry or western blot before release? If not, it is possible that Cas9 also induced DSBs in S/G2.

We thank reviewer #3 for requesting this control experiment. We now provide a Western blot of the Cas9-GFP protein in blocked pro-B cells (Figure S10A). The Cas9 protein level is decreased 48 hours post nucleofection that corresponds to the time we wash off the ABLki inhibitor and virtually undetectable 72 hours post nucleofection which corresponds to 24 hours after release of the cells (Figure 3C). 24 hours after release from the ABLki, less than 5% of the cells, typically 2 to 3%, are in S-phase (Figure 1B and Figure S3), thus we are confident that the vast majority – if not all – of the Cas9-induced DSBs occur in G1-arrested cells.

REVIEWERS' COMMENTS:

Reviewer #2 (Remarks to the Author):

I am happy with how the authors have addressed all points raised. The manuscript has further improved from its original version, which was already very accomplished. I would like to congratulate all the authors on this excellent study.

REVIEWERS' COMMENTS:

Reviewer #2 (Remarks to the Author):

I am happy with how the authors have addressed all points raised. The manuscript has further improved from its original version, which was already very accomplished. I would like to congratulate all the authors on this excellent study.

We thank the reviewer for his kind remarks and congratulations.